# Transcription of MERVL retrotransposons is required for preimplantation embryo development

Akihiko Sakashita[1,3], Tomohiro Kitano [1,3], Hirotsugu Ishizu[1], Youjia Guo [1], Harumi Masuda[1], Masaru Ariura[1], Kensaku Murano [1] & Haruhiko Siomi [1,2]✉

Zygotic genome activation (ZGA) is a critical postfertilization step that promotes totipotency and allows different cell fates to emerge in the developing embryo. MERVL (murine endogenous retrovirus-L) is transiently upregulated at the two-cell stage during ZGA. Although MERVL expression is widely used as a marker of totipotency, the role of this retrotransposon in mouse embryogenesis remains elusive. Here, we show that full-length MERVL transcripts, but not encoded retroviral proteins, are essential for accurate regulation of the host transcriptome and chromatin state during preimplantation development. Both knockdown and CRISPRi-based repression of MERVL result in embryonic lethality due to defects in differentiation and genomic stability. Furthermore, transcriptome and epigenome analysis revealed that loss of MERVL transcripts led to retention of an accessible chromatin state at, and aberrant expression of, a subset of two-cell-specific genes. Taken together, our results suggest a model in which an endogenous retrovirus plays a key role in regulating host cell fate potential.

Fertilization and early preimplantation development are processes in which unipotent gametes unite and acquire totipotency (Fig. 1a). After fertilization, embryos undergo zygotic genome activation (ZGA), a process that is widely conserved in vertebrates[1–3]. ZGA involves a transcriptional burst of hundreds to thousands of two-cell-specific genes. At this point, gene expression switches from a maternal to zygotic program[2,4]. ZGA occurs in two distinct waves called minor and major ZGA[5]. In mice, minor ZGA occurs from S phase in the zygote to G1 phase in the early two-cell stage embryo, whereas the major wave occurs during the second round of DNA replication at the middle-to-late two-cell stage[6,7]. Both waves of ZGA are critical for the embryo to acquire developmental competence[6,8]. However, the molecular events that drive ZGA and lead to acquisition of totipotency and developmental competence are still enigmatic.

Approximately 40% of the mouse genome is occupied by transposable elements (TEs), mobile genetic elements of which ~10% are endogenous retrovirus (ERV)[9]. Notably, the expression of murine endogenous retrovirus with leucine transfer RNA primer binding site (MERVL) is specifically activated at the two-cell stage concomitant with ZGA[10–12]. Recently, the transcription factor DUX, which is expressed during minor ZGA, was documented as an upstream regulator that activates two-cell genes and MERVL[13–15]. Furthermore, the MERVL long terminal repeat (LTR) promoter drives a subset of two-cell genes and generates chimeric transcripts with the host genes[12]. The above findings suggest that DUX/MERVL may activate an early transcriptional network that is required for ZGA and totipotency.

In 2012, Macfarlan et al. found that a rare transient cell population (~1%) in mouse embryonic stem cell (ESC) and induced pluripotent stem cell cultures expresses high levels of MERVL and two-cell genes without expression of pluripotent inner cell mass (ICM) maker genes, such as *Pou5f1* (also known as *Oct4*), *Sox2* and *Nanog*[12]. MERVL expression has been used as a marker for totipotent cells, as MERVL+ cells can commit to both embryonic and extraembryonic lineages after injection into recipient embryos at the eight-cell and morula stages[12,16,17].

[1]Department of Molecular Biology, Keio University School of Medicine, Tokyo, Japan. [2]Human Biology Microbiome Quantum Research Center (WPI-Bio2Q), Keio University, Tokyo, Japan. [3]These authors contributed equally: Akihiko Sakashita, Tomohiro Kitano. ✉e-mail: awa403@keio.jp

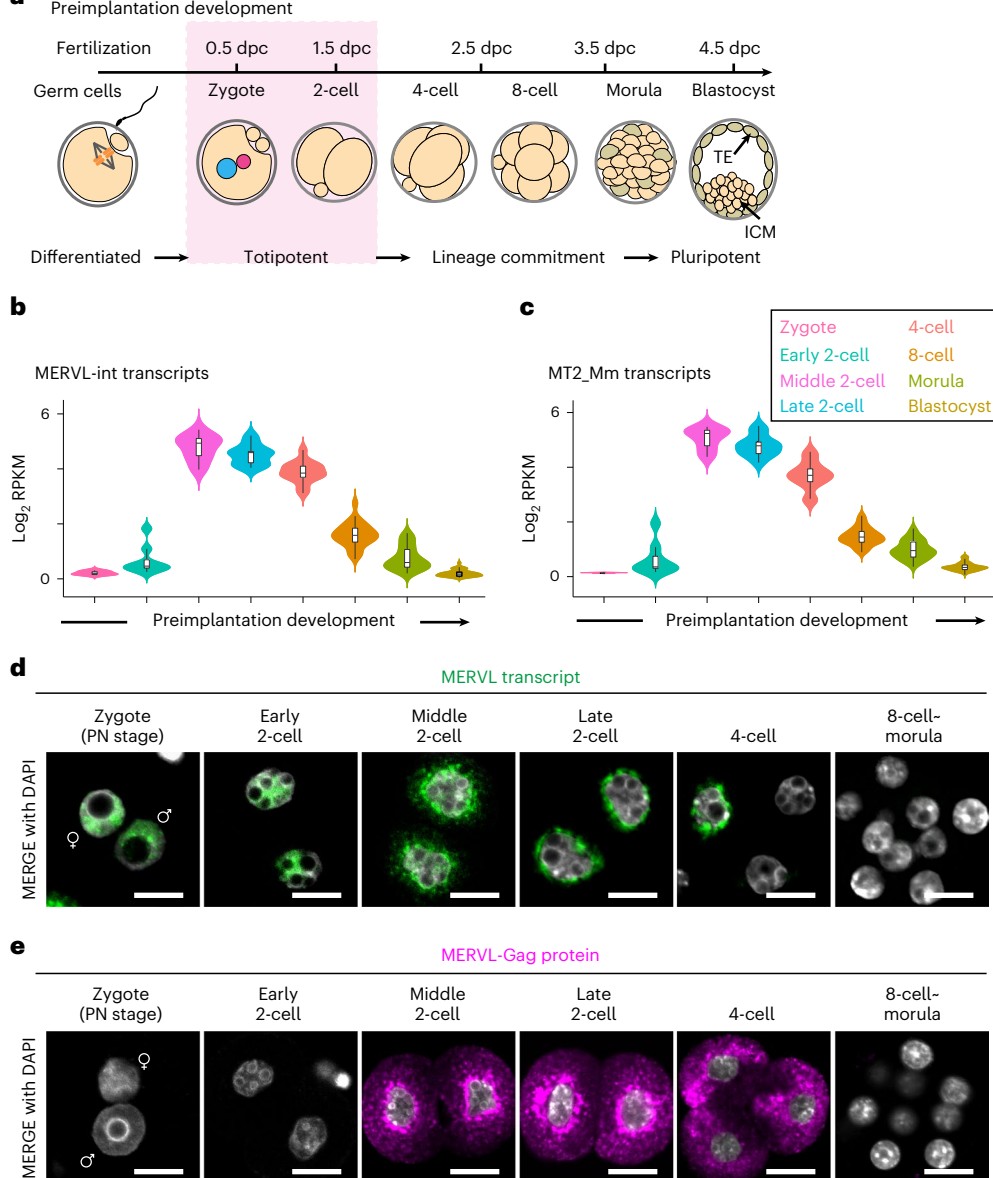

**Fig. 1 | MERVL RNA exhibits dynamic nuclear-cytoplasmic expression during early stages of mouse preimplantation development. a,** Schematic of mouse preimplantation development. Totipotency is restricted to early-stage development (that is, zygote and two-cell stages). Blastomeres gradually transition to a pluripotent state from the four-cell stage onward and develop into a blastocyst consisting of inner cell mass (ICM) and trophectoderm (TE) before implantation in the uterus at 4.5-days postcoitum (dpc). **b,c,** Violin plots showing the log$_2$-transformed reads per kilobase of exon per million reads mapped (log$_2$RPKM) values of MERVL-int (b) and its LTR promoter, MT2_Mm (c) during preimplantation development. Each plot encompasses box plot; central bars represent medians, box edges indicate 50% of data points and the whiskers show 90% of data points. **d,** Representative images of smFISH for MERVL RNA with 4,6-diamidino-2-phenylindole (DAPI) counterstain during preimplantation development, from four independent experiments. Scale bars, 20 μm. ♀, female pronucleus (PN); ♂, male PN. **e,** Representative images of immunofluorescence staining for MERVL-Gag protein with DAPI counterstain during preimplantation development, from six independent experiments. Scale bars, 20 μm. ♀, female PN; ♂, male PN. Data for panels in **b** and **c** are available as source data.

Despite the above findings, the function of MERVL itself remains unclear. Here, we overcome technical limitations in interrogating TE functions and analyze the role of MERVL in preimplantation development. We found that depletion of MERVL transcripts resulted in embryonic lethality due to defects in early lineage specification and genome stability, demonstrating that MERVL is essential for mouse preimplantation development.

## Results

### MERVL exhibits distinct localization in mouse embryos

To understand the dynamics of MERVL expression, we first analyzed publicly available single-cell RNA-sequencing (scRNA-seq) datasets

from each blastomere at eight representative stages of preimplantation development[18] (Fig. 1a). To define regions of nonredundant MERVLs in mouse genome, we used RepeatMasker to annotate the genome for unique interspersed internal regions of MERVL (MERVL-int, $n = 1,426$) and LTR promoters of the MERVL (MT2_Mm, $n = 2,366$). The expression of MERVL and its LTR promoter culminated in the middle of the two-cell stage and then gradually decreased until blastocyst stage (Fig. 1b,c).

We also set out to investigate the expression and localization of MERVL transcripts in preimplantation embryos using single-molecule fluorescence in situ hybridization (smFISH). Interestingly, smFISH revealed that MERVL expression is detectable in the nuclei from zygotes and early two-cell stage embryos in which polyadenylated MERVL

mRNA cannot be detected (Fig. 1b,d). Afterwards, MERVL RNA gradually translocated from the nucleus at middle two-cell stage onward and was highly restricted to the cytoplasm by late two-cell stage (Fig. 1d). These changes in MERVL transcript localization were consistent with increased MERVL protein levels during the middle two-cell stage (Fig. 1e). These observations raise the possibility that nuclear MERVL transcript has distinct roles in gene regulation in the early stages of preimplantation development compared to cytoplasmic MERVL transcript, leading us to investigate MERVL function further.

## MERVL-KD results in embryonic lethality

Inconsistencies regarding the early embryonic phenotypes of MERVL knockdown (KD) in previous studies[11,19,20], led us to re-examine the KD effects of MERVL on preimplantation development. To this end, we developed specific antisense oligonucleotides (ASOs) that target interspersed MERVL copies (Fig. 2a and Extended Data Fig. 1a,b). After predicting the genome-wide target sites of individual ASOs using BLASTn, we confirmed that 46.9% ($n = 669/1,426$) of MERVL copies were targeted by at least one ASO with up to two mismatches allowed (Extended Data Fig. 1c). Subsequently, we confirmed that our ASO sequences efficiently targeted full-length MERVL ($\geq$5 kb, n = 377/556, 67.8%), by combining three independent anti-MERVL ASOs (Extended Data Fig. 1d). We experimentally validated the KD efficiency of each ASO using a recently developed ESC-based in vitro system (Extended Data Fig. 1e and Methods)[21] in which MERVL expression was drastically reduced at both the mRNA and protein levels (Extended Data Fig. 1f–h). Injection of each ASO into the male pronucleus of zygotes also leads to substantial reduction of MERVL RNA signal at the late two-cell stage (Extended Data Fig. 2a). Because we noted that cocktail of three independent ASOs increased MERVL-KD efficiency (Fig. 2b and Extended Data Fig. 2), mixed ASOs (1:1:1 = 20 μM) were used in subsequent experiments.

Next, we monitored the effects of MERVL-KD using ASOs on preimplantation development (Fig. 2c,d). MERVL-KD embryos displayed a significant developmental delay from 2.5 dpc (Fig. 2d). Despite longer culture, a greater percentage of MERVL-KD embryos remained at the two- to four-cell stage (26.2% for control and 82.7% for MERVL-KD at 2.5 dpc (**$P = 0.00533$, chi-square test; Fig. 2d)). Strikingly, at 4.5 dpc, almost no blastocyst formation was observed in MERVL-KD embryos (Fig. 2c,d). These findings revealed that majority of MERVL-KD embryos suffered a developmental delay and halted development before blastocyst formation. To exclude the possibility that the reduction of MERVL transcript level and aberrant developmental phenotypes in MERVL-KD embryos might arise from unexpected artificial effects due to the presence of excessive small nucleotides, such as DNA replication stress and innate immune response, we performed microinjection of sense oligonucleotides (SOs), which are complementary to individual MERVL-targeting ASOs into the male pronucleus of zygotes.

This confirmed that injection of sense oligonucleotides (SOs) did not affect the expression of MERVL and preimplantation development (Extended Data Fig. 2).

Alternatively, we also tested KD of MERVL using CasRx, another KD strategy with a CRISPR-based RNA-targeting system[22] (Extended Data Fig. 3a). One day after induction of CasRx-mediated KD, no MERVL protein was detected in MERVL-KD two-cell stage embryos, and nuclear and cytoplasmic MERVL RNA signals remained, albeit with clear reductions compared to that of control (Extended Data Fig. 3b). Even with this KD inefficiency, the preimplantation development upon CasRx-mediated KD partially phenocopied that observed in ASO-mediated KD embryos (Extended Data Fig. 3c,d).

Of note, the phenotype of ASO-mediated MERVL-KD embryos is reminiscent of that observed in loss-of-function of genes associated with ICM and trophectoderm differentiation[23–27], implying that MERVL transcription might function in early cell lineage specification during preimplantation development. To test this hypothesis, we examined mRNA levels of genes related to ICM (*Oct4*, *Sox2* and *Nanog*) and trophectoderm differentiation (*Tead4*, *Tcfap2c* and *Cdx2*) in MERVL-KD embryos at 3.5 dpc, by RT-qPCR (Fig. 2e). *Sox2*, *Nanog* and *Cdx2* mRNA levels were significantly reduced in MERVL-KD morula, in contrast to *Oct4*, *Tead4* and *Tcfap2c* transcripts, which increased (Fig. 2e). These data suggest that MERVL transcription during early stages of preimplantation development is required for subsequent proper expression of genes linked to the earliest cell lineage specification events. To further investigate the molecular etiologies of the phenotype upon MERVL-KD, we examined OCT4 and CDX2 protein expression using immunofluorescence analysis at the morula stage (Fig. 2f). MERVL-KD morula is notably composed of significantly fewer blastomeres (Fig. 2g) with a significant reduction in OCT4 and CDX2 protein expression (Fig. 2h). Notably, MERVL-KD embryos also displayed increased hallmarks of genomic instability such as nuclear deformation and micronuclei, and apoptosis (Fig. 2i,j). In sum, we conclude that the depletion of MERVL transcript results in disruption of lineage specification, cell death, and ultimately early embryonic lethality.

## *Cis*-acting functions of MERVL is essential for development

We next interrogated which products from MERVL were required for preimplantation development. Firstly, we used two independent short interfering RNA (siRNAs) to knock down MERVL (Fig. 3a). Upon siRNA-mediated KD of MERVL, no cytoplasmic MERVL RNA and protein were detected in late two-cell stage embryos, whereas the nuclear signal of MERVL RNA was still intact in early two-cell stage embryos (Fig. 3b). Consequently, although the phenotypes of siRNA-mediated MERVL-KD are inconsistent between previous studies[19,20], we confirmed that siRNA-mediated KD of MERVL had almost no significant impact on preimplantation development (61.6% for control and 66.1% for MERVL-KD ($P = 0.5966$, chi-square test) (Fig. 3c,d)), suggesting that

**Fig. 2 | MERVL plays a critical role in early lineage specification and maintaining genomic stability of preimplantation embryos. a**, Schematic of full-length MERVL, indicating the positions of ASOs. **b**, Representative images of smFISH for MERVL RNA (green) with DAPI counterstain (gray) in control and MERVL-KD embryos at early and late two-cell stages, from six independent experiments (top). Representative images of immunofluorescence staining for MERVL-Gag protein (pink) with DAPI counterstain (gray) in control and MERVL-KD embryos at late two-cell stage, from three independent experiments (bottom). Scale bars, 20 μm. **c**, Representative phase-contrast images of 4.5 dpc blastocysts in control and MERVL-KD, from four independent experiments. Scale bars, 100 μm. **d**, Percentage of embryos by stages of development in control ($n = 84$) and MERVL-KD ($n = 104$). NS, not significant; **$P < 0.01$; ***$P < 0.001$, chi-square test. **e**, Expressions of ICM- and TE-associated genes, as measured by qRT-PCR in control and MERVL-KD morula at 3.5 dpc. Bars show means with standard error of the mean (s.e.m.). Dots represent biological replicates ($n = 6$). *$P < 0.05$, **$P < 0.01$, ***$P < 0.001$, two-tailed unpaired *t*-tests. **f**, Representative images of immunofluorescence staining for OCT4 and CDX2 with DAPI

counterstain in control and MERVL-KD morula at 3.5 dpc, from three independent experiments. Scale bars, 20 μm. **g,h**, Dot plots showing the total number of blastomere per embryo and relative intensity for OCT4 and CDX2 per nucleus in control and MERVL-KD morula at 3.5 dpc, normalized to DAPI signal. Central bars represent medians, and the top and bottom lines encompass 50% of the data points. ***$P < 0.001$, two-tailed unpaired *t*-tests. **i**, Representative images of immunofluorescence staining for E-cadherin (E-Cad, green) with DAPI counterstain (gray) in control and MERVL-KD morula at 3.5 dpc, from two independent experiments (left). Arrows and arrowheads indicate nuclear deformation and micronuclei. Scale bars, 20 μm. Bar chart showing the percentage of abnormal nuclear morphology at morula stage (right). *$P < 0.05$, chi-square test. **j**, Representative images of immunofluorescence staining for cleaved caspase-3 (cCasp3, green) with DAPI counterstain (gray) in control and MERVL-KD morula at 3.5 dpc, from three independent experiments (left). Scale bars, 20 μm. Bar chart showing the percentage of apoptotic cells per embryo at morula stage (right). ***$P < 0.001$, chi-square test. Data for panels in **d**, **e** and **g–j** are available as source data.

proteins encoded by MERVL are dispensable for preimplantation development.

To determine whether transcribed MERVL RNA plays a critical role in preimplantation development, we used a MERVL RNA construct with point mutations conferring resistance to ASO-mediated KD (Fig. 3e) and co-injected an ASO-resistant MERVL RNA with mixed ASOs

into zygotes. MERVL transcript was observed in nuclei of MERVL-KD early two-cell stage embryos at levels comparable to control embryos, when ASO-resistant MERVL transcript was present (Fig. 3f). However, upon injection of MERVL RNA, no cytoplasmic MERVL RNA signal was detected in late two-cell stage embryos. (Fig. 3f), which is to be expected, given that our exogenous MERVL construct is likely degraded

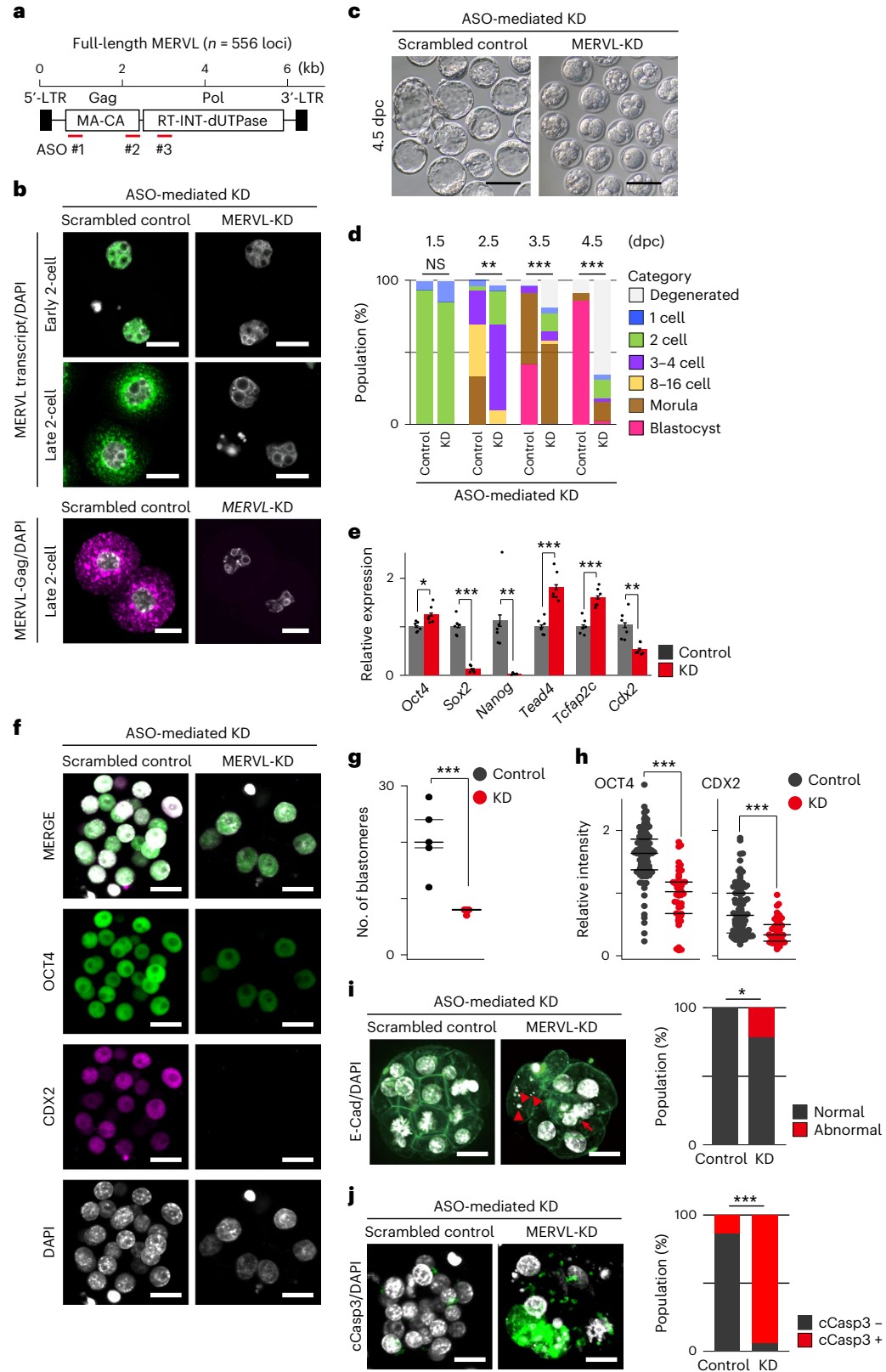

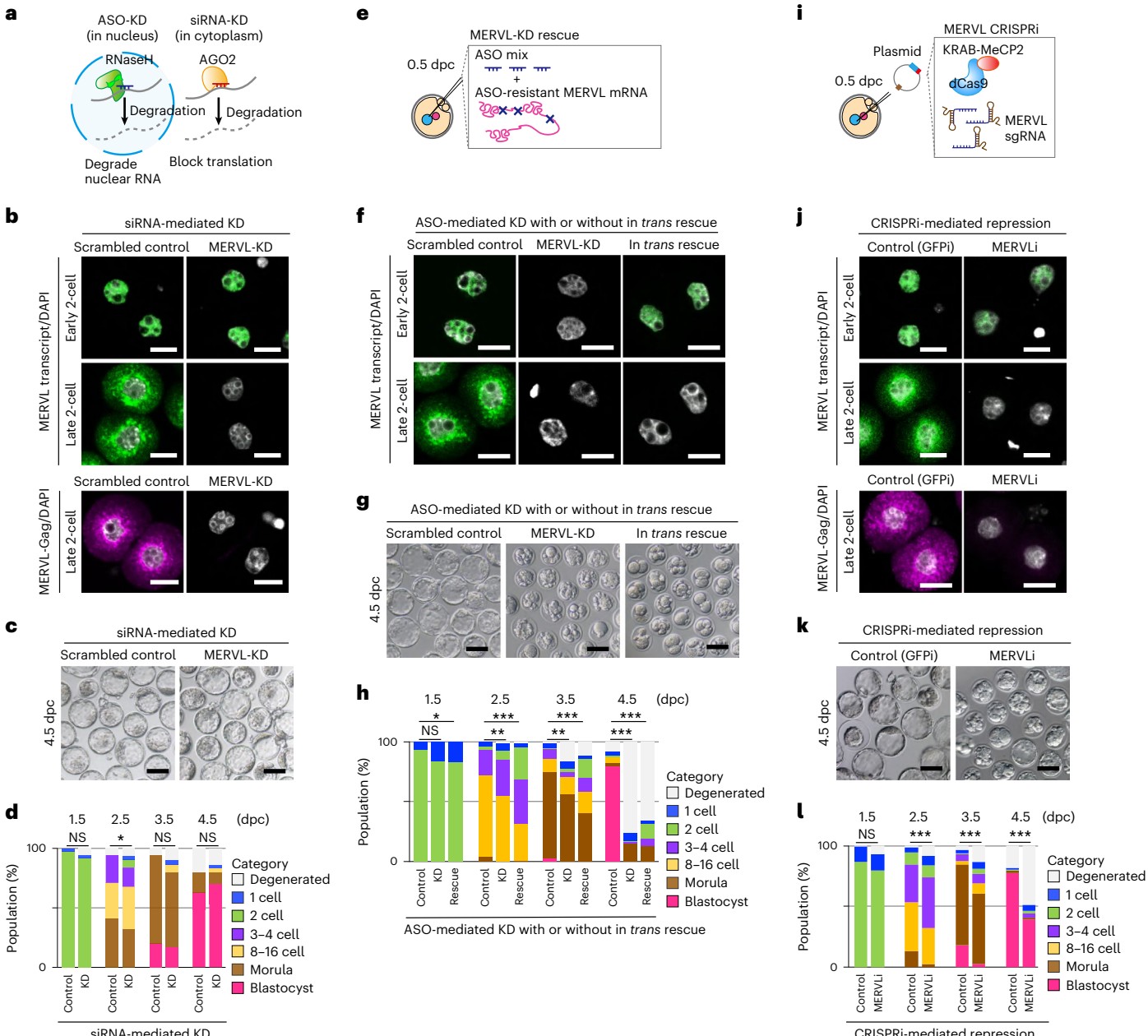

**Fig. 3 | Retroviral proteins and *trans*-acting MERVL RNA are dispensable for preimplantation development. a**, Schematic of the distinct mechanisms underlying ASO- versus siRNA-mediated RNA targeting. **b** Representative images of smFISH for MERVL RNA (green) with DAPI counterstain (gray) in control and MERVL-KD embryos at early and late two-cell stages, from four independent experiments (top). Representative images of immunofluorescence staining for MERVL-Gag protein (pink) with DAPI (gray) counterstain in control and MERVL-KD embryos at late two-cell stage, from three independent experiments (bottom). Scale bars, 20 μm. **c**, Representative phase-contrast images of 4.5 dpc blastocysts in control and MERVL-KD by siRNA, from three independent experiments. Scale bars, 100 μm. **d**, Percentage of embryos by stages of development in control (*n* = 73) and MERVL-KD (*n* = 59). **P* < 0.05, chi-square test. **e**, Schematic of experimental procedure for in *trans* rescuing MERVL-KD by ASOs. **f**, Representative images of smFISH for MERVL RNA with DAPI counterstain in early and late two-cell-stage embryos in each experimental condition, from four independent experiments. Scale bars, 20 μm. **g**, Representative phase-

contrast images of 4.5 dpc blastocysts in each experimental condition, from two independent experiments. Scale bars, 100 μm. **h**, Percentage of embryos by stages of development in each experimental condition (*n* = 78 for control, *n* = 80 for MERVL-KD and *n* = 132 for in *trans* rescue). **P* < 0.05; ***P* < 0.01; ****P* < 0.001, chi-square test. **i**, Schematic of experimental procedure for CRISPRi targeting the MERVL sequence. **j**, Representative images of smFISH for MERVL RNA (green) with DAPI counterstain (gray) in GFPi control (CRISPRi targeting GFP) and MERVLi (CRISPRi targeting MERVL) embryos at early and late two-cell stages, from three independent experiments (top). Representative images of immunofluorescence staining for MERVL-Gag protein (pink) with DAPI counterstain (gray) in GFPi control and MERVLi embryos at late two-cell stage, from three independent experiments (bottom). Scale bars, 20 μm. **k**, Representative phase-contrast images of 4.5 dpc blastocysts upon GFPi control and MERVLi condition, from 3 independent experiments. Scale bars, 100 μm. **l** Percentage of embryos by stages of development upon GFPi control (*n* = 149) and MERVLi (*n* = 137). ****P* < 0.001, chi-square test. Data for panels in **d**, **h** and **l** are available as source data.

by RNA surveillance pathways[28]. Although nuclear MERVL transcript was present in MERVL-KD embryos by co-injection with ASO-resistant MERVL RNA, the embryonic lethality was not rescued (Fig. 3g,h).

Collectively, our data demonstrated that neither encoded proteins nor *trans*-acting RNA of MERVL is indispensable for preimplantation development.

To gain better insight into *cis*-acting functions of MERVL, we employed the CRISPR interference (CRISPRi) system[29–31] (Fig. 3i). After microinjection, we verified that the expression of MERVL protein was efficiently suppressed in two-cell stage embryos upon CRISPRi to MERVL (MERVLi), whereas MERVL transcriptional level was significantly decreased in MERVLi embryos compared to that in GFPi control, but not fully silenced (Fig. 3j). Notably, blastocyst formation was severely impaired upon MERVLi (Fig. 3k). Only 38.7% MERVLi embryos reached blastocyst stage, compared with 78.5% for GFPi control embryos (Fig. 3l). These findings showed the necessity of MERVL transcription in early stage of development at the chromatin level.

## A subset of two-cell genes are dysregulated in MERVL-KD embryos

To assess the effects of ASO-mediated MERVL-KD on ZGA, we made use of 5′-ethynyluridine (EU) to assay global transcriptional levels in control and MERVL-KD two-cell stage embryos (Fig. 4a). We observed no significant change in zygotic transcriptional activity between control and MERVL-KD two-cell stage embryos during major ZGA (Fig. 4b,c).

Further, we carried out total RNA-seq analyses in control and MERVL-KD embryos (Extended Data Fig. 4a,b). The RNA-seq dataset provided accurate gene expression profiles (Fig. 4d and Supplementary Tables 1 and 2). When we compared differentially expressed genes (DEGs) between control and MERVL-KD embryos, we found that 938, 745 and 277 genes were significantly dysregulated in MERVL-KD embryos, at the two-cell, four-cell and eight-cell stages, respectively (Fig. 4e, Supplementary Table 3). Of these DEGs, only 3–15% of genes were defined as 'maternally inherited RNA' in a database of transcriptome in mouse early embryos version 2 (DBTMEE v2 (refs. 32,33)) across all stages examined (Fig. 4f), suggesting that MERVL-KD led to major defects in expression of zygotic genes rather than maternal mRNA clearance. In contrast to annotated genes, among all interspersed repetitive elements, only MERVL and a small number of closely related subtypes were significantly downregulated in MERVL-KD preimplantation embryos (Extended Data Fig. 5a,b). Moreover, largely consistent with previous reports[12], we also detected 77 genes that generated chimeric transcripts with junction to MT2_Mm in both control and MERVL-KD embryos. However, the overall expression level of chimeric transcripts is unchanged in MERVL-KD embryos (Extended Data Fig. 5c).

To better understand the functions of the DEGs, we performed gene ontology (GO) and ChIP Enrichment Analysis (ChEA)[34], indicating that TP53 (also known as p53)-related terms were significantly enriched in each set of upregulated DEGs (Fig. 4g and Extended Data Fig. 4c). Taken together, these transcriptome analyses raise the possibility that activated p53 ectopically accumulates in the nuclei of MERVL-KD embryos. Because DNA damage and activated p53 contribute to the induction of two-cell genes in early-stage preimplantation embryos and ESCs[35], we reasoned that the genes significantly upregulated in

MERVL-KD embryos might be enriched for two-cell and p53-target genes. To test this hypothesis, we next compared DEGs with the lists of two-cell genes (as defined in Macfarlan et al.[12]) and DBTMEE v2 (refs. 32,33). Significant enrichment of two-cell genes was observed in all DEG sets ($P < 0.001$, hypergeometric test for overrepresentation; Fig. 4h). Strikingly, upregulated DEGs in MERVL-KD four-cell embryos exhibited the highest increase in enrichment of two-cell genes (Fig. 4h). Unsupervised hierarchical clustering of all two-cell gene transcripts revealed that more than half of two-cell genes were dysregulated in at least one developmental stage of MERVL-KD embryos (Extended Data Fig. 4d). In addition, two-cell- and four-cell-transient genes were also enriched in all sets of upregulated DEGs (Fig. 4h). These findings argue that MERVL-KD embryos still retain a two-cell/totipotent-like transcriptome even at mid-preimplantation stages (that is, four-cell and eight-cell stages). Indeed, representative track views demonstrated that two-cell gene loci abnormally maintained high expression levels of two-cell genes in MERVL-KD embryos (Fig. 4i). Consistent with our hypothesis, we also observed a significant increase in phospho-S15-p53 signal intensity in MERVL-KD embryos compared to that in controls (Fig. 4j,k). Overall, these results indicate that a subset of two-cell genes are persistently active due to defects in MERVL transcription.

We also performed total RNA-seq analysis for GFPi control and MERVLi embryos (Supplementary Table 1). Hierarchical clustering and Pearson correlation analyses for the expression of all annotated transcripts showed higher similarity between ASO-mediated MERVL-KD and MERVLi embryos (Extended Data Fig. 6a,b), suggesting that the extensive transcriptional anomalies in MERVLi embryos at least partially resemble that in MERVL-KD embryos. Indeed, although only moderate reduction of MERVL expression was observed in MERVLi two-cell stage embryos due to limited effectiveness of CRISPRi for depleting nuclear MERVL RNA (Extended Data Fig. 6c), we identified 872, 535 and 273 genes that were differentially expressed in MERVLi embryos, at the two-cell, four-cell and eight-cell stages, respectively (Extended Data Fig. 6d and Supplementary Table 4). Importantly, significant enrichment of two-cell genes was observed in all sets of DEGs through MERVLi preimplantation development ($P < 0.001$, hypergeometric test for overrepresentation; Extended Data Fig. 6e). Consistently, a representative two-cell gene maintained abnormally high expression level in MERVLi embryos (Extended Data Fig. 6f). These profiles demonstrated that MERVLi embryos exhibited a transcriptome that was highly similar to that of MERVL-KD embryos, corroborating that the embryonic phenotypes in MERVL-KD and MERVLi embryos arise from defects in *cis*-regulatory function of MERVL.

## MERVL modulates dynamic changes in accessible chromatin

To further investigate the correlation between transcription and chromatin state, we next asked whether MERVL-KD impacts chromatin accessibility at the transcription start sites (TSSs) of dysregulated two-cell

**Fig. 4 | Ablation of MERVL impacts the expression of a subset of two-cell genes through preimplantation development. a**, Schematic of timing for minor and major ZGA (adapted from Abe et al.[6]). **b**, Representative images of EU incorporation assay (green) with Hoechst 33342 counterstain (gray) in control and MERVL-KD two-cell stage embryos, from three independent experiments. Scale bars, 20 μm. **c**, The relative intensity of EU per nucleus, normalized to Hoechst 33342 signal. NS, not significant by two-tailed unpaired *t*-tests. **d**, Bidimensional principal-component (PC) analysis of gene expression profiles in control and MERVL-KD embryos through preimplantation development. **e**, RNA-seq differential gene expression analysis: MERVL-KD versus control embryos obtained at two-cell, four-cell and eight-cell stages; 938, 745 and 277 genes evinced significant changes in expression in MERVL-KD embryos (blue circle, fold change ≥ |2|, *P* value adjusted (*P*adj) < 0.05, binomial test with Benjamini–Hochberg correction). **f**, Stacked bar chart shows the number of upregulated (up)- and downregulated (down)-DEGs. The populations of maternally inherited RNA (as defined in DBTMEE v2; refs. 32,33) and others (zygotically expressed genes) are highlighted in red and blue. **g**, Predicted factors that are upstream of all up-DEGs upon MERVL-KD, assessed

by the ChEA database[34]. ChIP-seq, chromatin immunoprecipitation with sequencing; HaCaT, cultured human keratinocyte; MEF, mouse embryonic fibroblast; MESC, mouse embryonic stem cell; LNCaP, lymph node carcinoma of the prostate. **h**, Bubble plot showing overlap between all DEGs in MERVL-KD embryos with the list of two-cell (2C) genes and DBTMEE v2 transcriptome categories. The bubble plot sizes show the $-\log_{10}[P \text{ values}]$ derived from a hypergeometric test. **i**, Track views show RNA-seq signals in control and MERVL-KD embryos, on two representative 2C gene loci (as defined in Macfarlan et al.[12]). The *y*-axis represents normalized tag counts for total RNA-seq in each sample. Refgene, refseq gene in NCBI and UCSC. **j**, Representative images of immunofluorescence staining for phosphorylated p53 on serine 15 (p-S15-p53, pink) with DAPI counterstain (gray) in control and ASO-mediated MERVL-KD embryos, from three independent experiments. Scale bars, 20 μm. **k**, Dot plots showing the relative intensity for p-S15-p53 per nucleus in control and MERVL-KD embryos, normalized to DAPI signal. Central bars represent medians, the top and bottom lines encompass 50% of the data points. *$P < 0.05$ and ***$P < 0.001$, two-tailed unpaired *t*-tests. Data for panels in **c**–**h** and **k** are available as source data.

genes by using the optimized low-input assay for transposase-accessible chromatin with sequencing (miniATAC-seq) method (Supplementary Table 1 and Extended Data Fig. 7)[36]. The overall enrichment pattern of

ATAC-seq signals at 10-kb intervals across the genome was modestly changed in MERVL-KD embryos during preimplantation development (Fig. 5a). Meanwhile, open chromatin at MERVL loci was significantly

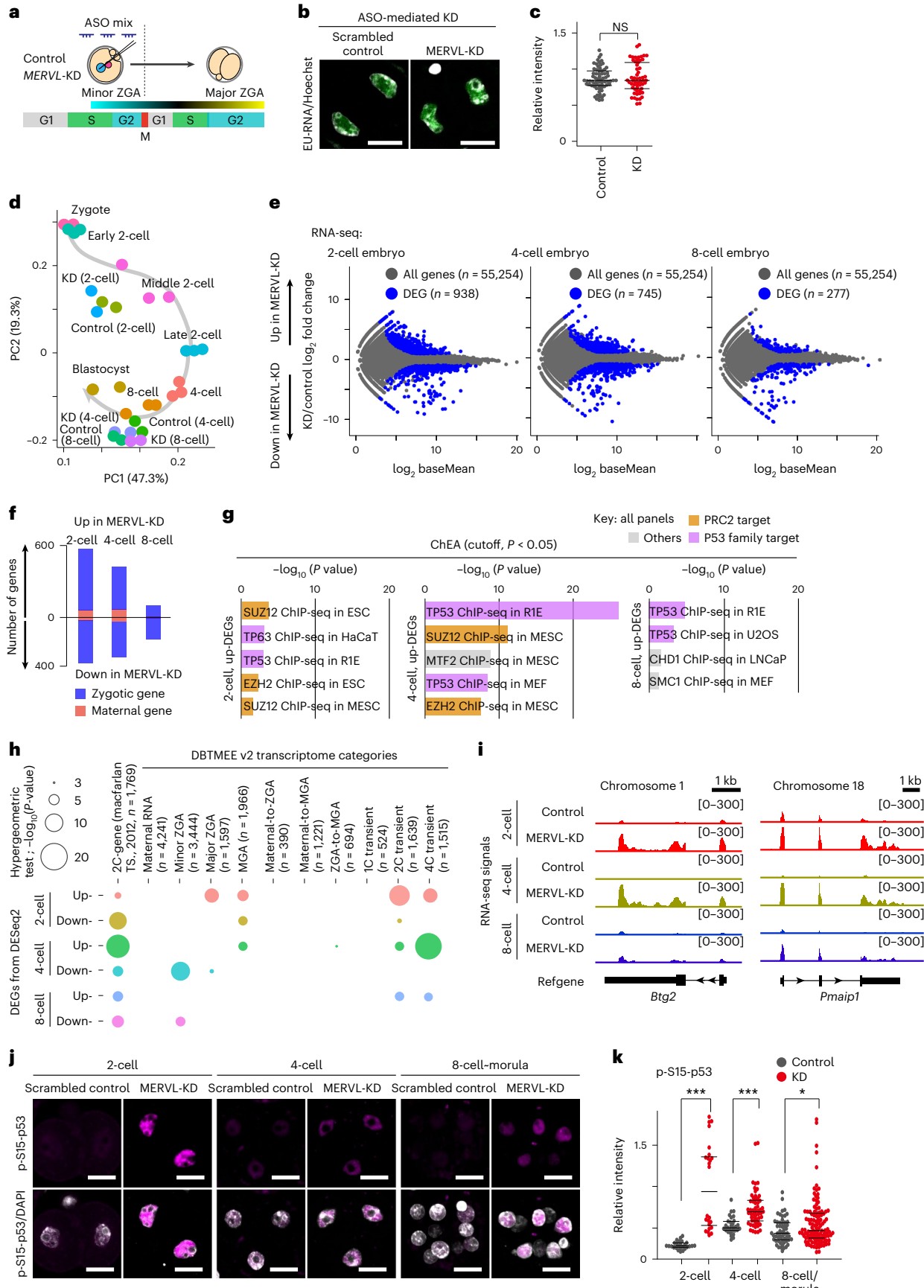

reduced in MERVL-KD two-cell-stage embryos (Fig. 5b). Because ASO-mediated KD might trigger premature transcriptional termination via degradation of the residual Pol II-associated transcripts[37], suggesting that the reduction of chromatin openness at MERVL loci is associated with premature termination of MERVL transcripts induced by KD.

Next, we determined whether the transcriptional changes in MERVL-KD embryos were associated with their chromatin states by examining enrichment of ATAC-seq signals at TSSs of two-cell genes, ATAC-seq signals showed significantly higher enrichment at the TSSs of two-cell genes in MERVL-KD embryos at the four-cell stage compared to control embryos (Fig. 5c). In line with this finding, a representative track view showed that an accessible chromatin state was detected at the TSS of a two-cell gene. This open state persisted into the mid-preimplantation stage upon MERVL-KD, a time when this locus is repressed/closed in control embryos (Fig. 5d). These results confirm that ectopic expression of two-cell genes in mid-preimplantation embryos upon MERVL-KD is associated with accessible chromatin states at these loci.

Furthermore, k-means clustering of ATAC-seq signals across all peak regions in control embryos yielded five clusters that fall into two distinct categories. Category A became accessible gradually as development proceeds (composed of cluster 1 and 2), and category B became specifically accessible in two-cell-stage embryos (composed of cluster 3–5) (Fig. 5e). Among each cluster, approximately half of ATAC-seq peaks were enriched at largely genic regions, including promoter, exon and intronic regions (Fig. 5f). GO analysis revealed that genes adjacent to peaks in category A were highly enriched for roles in 'chromatin organization' and 'posttranscriptional regulation', such as those involved in regulation of the pluripotent state (Fig. 5g, top)[38,39]. Moreover, the peaks in category A contained genic regions that associated with pluripotency, including *Oct4* and *Cdx2* (Fig. 5g, top, and Supplementary Table 5). Conversely, the peaks in category B were characterized by genes and biological terms associated with totipotent and two-cell states, such as *Dux* and *Zscan4c*, whose expression was restricted to two-cell stage embryos (Fig. 5g, bottom, and Supplementary Table 5)[13,14,40]. Finally, we examined ATAC-seq signals in each cluster and compared control and MERVL-KD embryos. In category B (cluster 3–5), analysis of average tag densities of ATAC-seq signals confirmed that chromatin accessibility was significantly decreased in peaks of both cluster 3 and 4 in MERVL-KD two-cell stage embryos (Fig. 5h). In contrast, from the four-cell stage onward, ATAC-seq signals were significantly increased in peaks of cluster 4 and 5 in MERVL-KD embryos (Fig. 5h), suggesting that a two-cell-like chromatin state was partially retained during preimplantation development of MERVL-KD embryos.

### MERVL-KD exhibits less abundant transcripts adjacent to MERVL

We interrogated the *cis*-regulatory functions of MERVL. First, we compared the expression levels of annotated genes (in GENCODE vM25) and

their distances to nearby full-length MERVL in control and MERVL-KD two-cell stage embryos. No significant differences were observed in the expression of adjacent annotated genes to MERVL, ranging from 0 to 500 kb (Fig. 6a). Intriguingly, in contrast to annotated genes, the unannotated transcripts from the adjacent regions to MERVL were significantly downregulated in MERVL-KD embryos, and this was positively correlated with their proximity to the full-length MERVL (Fig. 6b).

To evaluate the expression levels of unannotated intergenic transcripts in control and MERVL-KD two-cell stage embryos, we used groHMM, a two-state hidden Markov model (HMM)-based algorithm[41]. This framework classified transcriptionally active regions (TARs) across the genome into annotated TAR (aTAR) that overlapped with existing gene annotation and unannotated TAR (uTAR) that did not overlap with regions of any genes, pseudogenes and MERVL families annotated by GENCODE and RepeatMasker (Fig. 6c), identifying in total 240,869 uTARs (Fig. 6d and Supplementary Table 6). Despite the relative scarcity of uTAR, which accounted for 27.5–34.6% of aligned reads, we detected 3,499 differentially expressed uTARs between control and MERVL-KD embryos (Fig. 6d and Supplementary Table 7). In particular, the vast majority of them were shown to be downregulated upon the loss of MERVL transcription (Fig. 6e). Representative track views demonstrated that transcriptional activities in MERVL-adjacent regions were significantly reduced at upstream of genic TSS and gene desert regions, in MERVL-KD two-cell embryos (Fig. 6f). Specifically, we found a higher fraction of differentially expressed uTARs preferentially located in the neighborhood of MERVL, compared with all called TARs (Fig. 6g). These data suggest a *cis*-regulatory function for MERVL in the regulation of unannotated transcripts from their adjacent loci. Moreover, concomitantly with the change in expression of uTARs, the intensity of ATAC-seq signal on differentially expressed uTARs was also decreased in MERVL-KD embryos (Fig. 6h), leading to the suggestion that MERVL-driven unannotated transcripts may play a role in chromatin remodeling.

## Discussion

We provided functional evidence that transcriptional activation of MERVL is essential for progression of development in mouse preimplantation embryos. Depletion of MERVL transcripts results in embryonic lethality with profound defects in development and is associated with dysregulation of MERVL including their adjacent transcripts, and retaining two-cell-like transcriptome and chromatin state (Fig. 6i). These findings suggest the possibility that MERVL transcription in totipotent cells may act as a switch for the transition from totipotency to pluripotency and is responsible for the onset of differentiation and ontogeny (Fig. 6i).

One of most striking consequences of loss of the MERVL transcript was complete defects in preimplantation development. Several groups have used KD strategies in an effort to characterize the role of MERVL in preimplantation development. However, results

---

**Fig. 5 | MERVL-KD embryos retain two-cell-like chromatin accessibility even at mid-preimplantation stage. a**, Genome-wide correlation of ATAC-seq signals by stages of development between control and MERVL-KD embryos. Enrichment levels per 10-kb bin are shown in log$_2$RPKM values. The Pearson correlation coefficient values (*R*) indicate the similarity between control and MERVL-KD embryos. **b**, Average tag density plots of ATAC-seq enrichment around MERVL-int copies (±5 kb) by stages of development in control and MERVL-KD embryos. RPM, reads per million. **Padj < 0.01, Mann–Whitney U test with Bonferroni correction. **c**, Average tag density plots of ATAC-seq enrichment around TSS (±1 kb) of 2C genes by stages of development in control and MERVL-KD embryos. **Padj < 0.01, Mann–Whitney U test with Bonferroni correction. **d**, Track view shows ATAC-seq signals in control and MERVL-KD embryos at two-cell, four-cell and eight-cell stages, on representative 2C gene locus (as defined in Macfarlan et al.[12]). The y-axis represents normalized tag counts for ATAC-seq in each sample. The flanking region around TSS is highlighted in red. **e**, Heatmaps showing ATAC-seq signals across all peak regions ±1 kb in control embryos at two-cell, four-cell, and eight-cell stages. Each peak was ordered by k-means clustering of ATAC-seq signal and yielded five clusters. Cluster 1–2, increased accessibility during preimplantation development (defined as category A); cluster 3–5, were transiently accessible at two-cell stage and reduced accessibility during preimplantation development (defined as category B). **f**, Stacked bar chart shows ATAC-seq peak distributions across genomic entities (intergenic, intron, exon and promoter) in each cluster. **g**, GO analysis of genes adjacent to ATAC-seq peaks of cluster 1–2 (category A) and cluster 3–5 (category B). The pluripotent-related and 2C genes were obviously enriched in gene set adjacent to category A and category B, respectively. **h**, Average tag density plots of ATAC-seq enrichment around peak regions (±1 kb) in each cluster by stages of development in control and MERVL-KD embryos. *Padj < 0.05, **Padj < 0.01, ***Padj < 0.001, Mann–Whitney U test with Bonferroni correction. Data for panels in **a**, **f** and **g** are available as source data.

from these studies were inconsistent. By performing liposome-based transfection with ASO against MERVL, Kigami et al. reported that MERVL-KD embryos displayed a ~50% decrease in the developmental

competence to the four-cell stage but no significant impact on developmental rate to the morula and blastocyst stages[11]. On the other hand, Huang et al. recently performed siRNA-mediated KD against

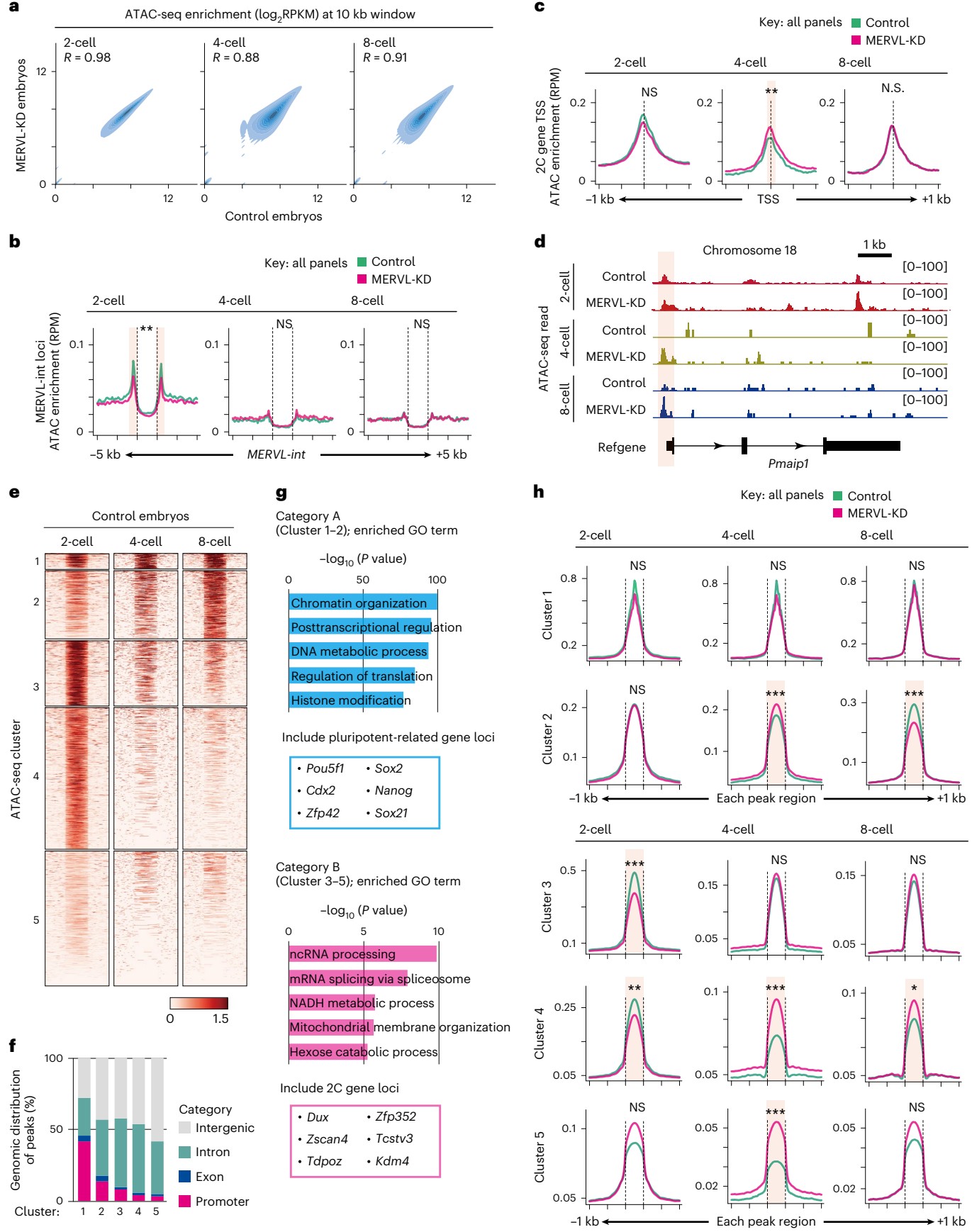

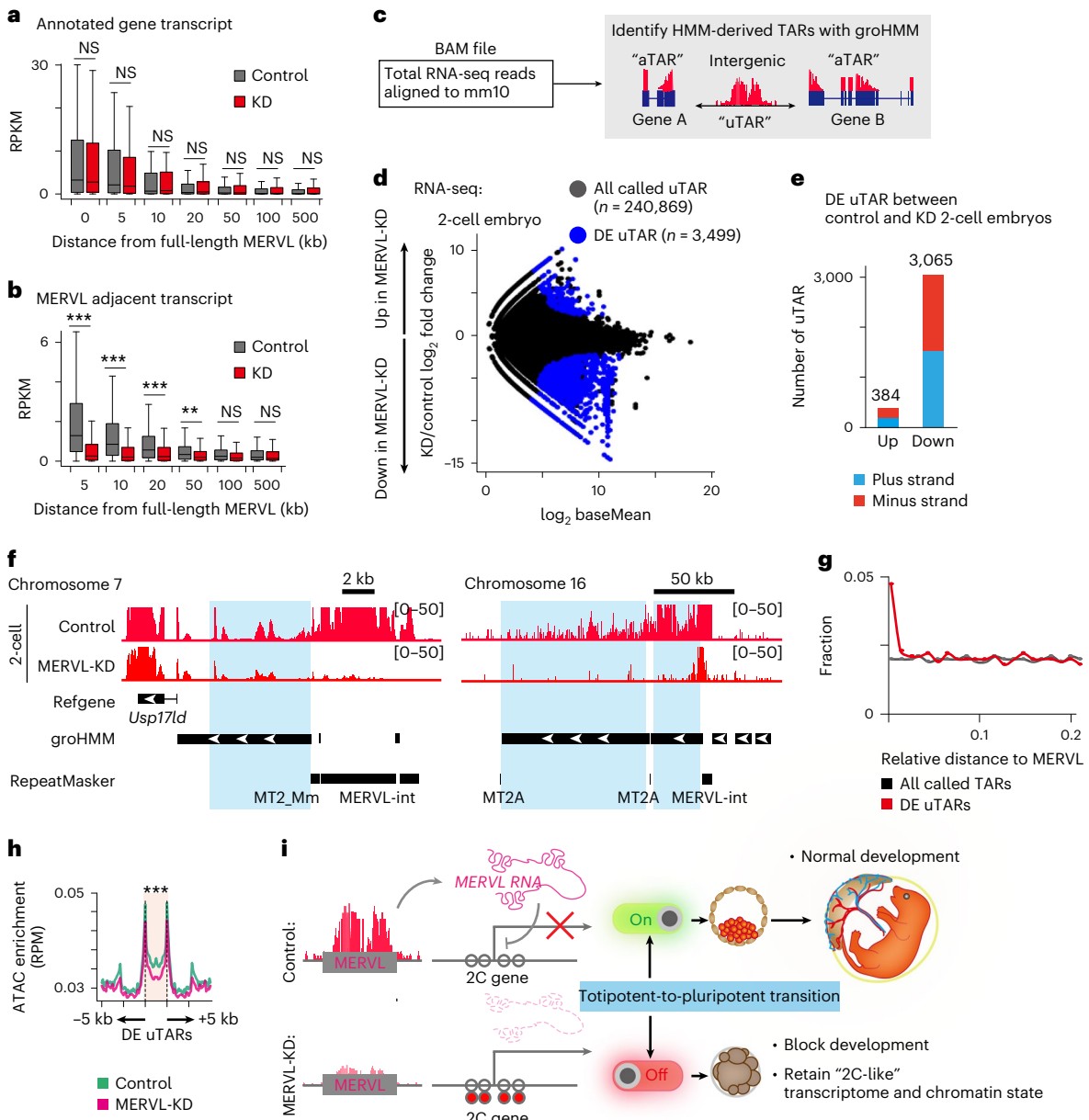

**Fig. 6 | Intergenic transcription from the adjacent regions to MERVL was interfered with by MERVL-KD. a**, Box plot showing the RPKM values for annotated genes, grouped by their distance to ASO-targeted full-length MERVL. Central bars represent medians, the boxes encompass 50% of the data points and the whiskers indicate 90% of the data points. NS, not significant, two-tailed unpaired *t*-tests. **b**, Box plot showing the RPKM values for adjacent transcripts grouped by their distance to ASO-targeted full-length MERVL. Central bars represent medians, the boxes encompass 50% of the data points and the whiskers indicate 90% of the data points. \*\**P* < 0.01, \*\*\**P* < 0.001, two-tailed unpaired *t*-tests. **c**, Schematic of procedure for transcript calling with groHMM, a two-state HMM-based algorithm[41]. aTAR, annotated transcriptionally active regions (blue rectangles); uTAR, unannotated intergenic transcriptionally active region (two-headed black arrow). **d**, RNA-seq differential expression analysis of uTARs between control and MERVL-KD two-cell stage embryos. MA plot showing differentially expressed (DE) uTARs between control and MERVL-KD embryos. DE uTARs were defined with a *P*adj < 0.01 (binomial test with Benjamini–Hochberg correction) and shown in blue circles (*n* = 3,499). **e**, Number of upregulated (up)-

and downregulated (down)-uTARs in MERVL-KD two-cell stage embryos from d. Each transcriptional direction in the DE uTARs is shown in blue (plus strand) and red (minus strand). **f**, Track views show RNA-seq signals in control and MERVL-KD two-cell stage embryos on two representative uTARs. DE uTARs are highlighted in blue. The *y*-axis represents normalized tag counts for total RNA-seq in each sample. **g**, Distributions of relative distances of DE uTARs (red) and all called TARs (black) to MERVL loci. **h**, Average tag density plots of ATAC-seq enrichment around DE uTARs (±5 kb) in control and MERVL-KD two-cell stage embryos. RPM, reads per million. \*\*\**P*adj < 0.001, Mann–Whitney *U*-test with Bonferroni correction. **i**, Model: MERVL-mediated totipotent-to-pluripotent transition during preimplantation development. MERVL and adjacent intergenic regions are expressed in a zygote and two-cell-stage-specific manner and suppresses a subset of 2C gene expressions through an unknown mechanism to facilitate the totipotent-to-pluripotent transition. KD of MERVL transcript compromises these processes and results in retention of two-cell-like states and embryonic lethality. Data for panels **a**, **b**, **d**, **e** and **g** are available as source data.

MERVL by microinjection of siRNA into the cytoplasm of zygotes[19]. siRNA-mediated KD of MERVL exhibited a mild developmental delay at four- to eight-cell stage, but no statistically significant differences were

observed on developmental rate between control and KD embryos[20]. In sharp contrast, ASO-mediated KD of MERVL in this study results in embryonic lethality with severe defects in first lineage specification

and genomic stability. The phenotypic differences appear to be explained by methodologies for KD. First, we microinjected multiple ASOs with 2′-O-methoxyethyl modifications, providing enhanced duplex stability and nuclease resistance[42], into zygotes resulting in efficient degradation of nuclear MERVL transcripts at much higher levels than liposome-based transfection (Figs. 2b and 3f). In addition, it is noteworthy that our ASOs more efficiently target interspersed full-length MERVL elements than that in previous study ($n = 377/556$, 67.8% versus $n = 259/556$, 46.6%). Consistent with previous studies, we confirmed that siRNA-mediated KD of MERVL did not affect development to the blastocyst stage (Fig. 3a–d), indicating that nuclear MERVL transcripts, but not encoded proteins, are required for preimplantation development. Although the activation of MT2_Mm inducing the two-cell state in previous study is readily appreciable[12,16,43], we also found that the expression levels of chimeric transcripts with MT2_Mm were unchanged in MERVL-KD two-cell stage embryos (Extended Data Fig. 5c), suggesting that the phenotypes of MERVL-KD embryos are unlikely to arise from dysregulation of chimeric transcripts with MT2_Mm. Given the results with siRNA-KD and in *trans* RNA rescue, *cis*-regulatory activity of MERVL loci is likely to play a key role in host chromatin remodeling at totipotent-to-pluripotent transition. As a proof-of-concept for this hypothesis, we performed CRISPRi-mediated repression of MERVL and revealed that MERVLi embryos partially mimicked ASO-mediated KD of MERVL with regard to developmental defects and retaining two-cell transcriptome (Fig. 3i–l and Extended Data Fig. 6).

Our findings showed that transcription itself and/or *cis*-acting RNA from MERVL loci is likely to play a critical role for host preimplantation development. Of note, having observed a subset of two-cell genes retaining expression in MERVL-KD mid-preimplantation embryos (Fig. 4g–i), MERVL transcripts may act to repress expression of two-cell genes during preimplantation development, at least in part. Several lines of evidence allow us to predict the mechanism by which MERVL may regulate host transcriptome and chromatin state in preimplantation development. First, a recent study demonstrated that MERVL drives the 3D reorganization of the genome in two-cell-like cells, the rare cell population in ESC culture, and early mouse embryos by providing a topologically associating domain boundary that is coupled to directional transcription from MERVL[44,45]. Thus, it is plausible that transcription itself from MERVL might be responsible for remodeling of chromatin structure during the transition from totipotency to pluripotency. As shown in Fig. 6, it is noteworthy that loss of MERVL transcription also leads to the reduction of transcription levels and chromatin accessibility in unannotated adjacent regions ranging up to 50 kb away. One possible explanation for this finding is that these cryptic transcripts may result from readthrough of RNA polymerase beyond MERVL elements, which leads to pervasive epigenetic changes. Related to our findings, Jachowicz et al. demonstrated that transcriptional activity of LINE1 that peaks at two-cell stage embryos affects global chromatin accessibility and unfolding[46]. Likewise, there has been reported that the transcribed RNA from other type of endogenous retroviruses (that is, intracisternal A-particles and MMERVK10C) can act in *cis* in ESC[47]. Accordingly, we speculate that en masse transcription of MERVL loci acts in *cis* as starting events for long-range 3D chromatin organization and remodeling toward the onset of ontogeny. Future studies can now interrogate the intriguing mechanisms that may underlie the important role of MERVL transcript in regulating the host genome that we have uncovered here.

## Online content

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

## Methods

### Animals

Eight-week-old female and male B6D2F1 (BDF1) mice were purchased from the Japan SLC. All animal experiments were reviewed and approved by the Institutional Animal Care and Use Committee (protocols 09105-(10) and 11045-(6)) at Keio University, where mice were maintained and fed ad libitum with standard diet and water in a temperature-, humidity- and light-controlled room (23°C ± 3°C, a humidity of 50% ±10% and 14-h light/10-h dark cycle).

### Cell culture

Doxycycline (Dox)-inducible Dux ESC line was generated using the PiggyBac Transposon System in our laboratory (hereafter referred to as ESC[DUX]), in which *Dux* expression was induced in the presence of Dox[21]. The ESC[DUX] was cultured in ESC medium (10% FBS, 1× sodium pyruvate, 1× GlutaMAX, 1× MEM non-essential amino acids solution, 1× penicillin/streptomycin and 0.055 mM β-mercaptoethanol in DMEM high glucose (4.5 g liter$^{-1}$)) containing 2i (1 μM PD0325901, LC Laboratories; and 3 μM CHIR99021, LC Laboratories) and LIF (1,000 U ml$^{-1}$, in-house) on cell-culture plates coated with 0.1% gelatin under feeder-free conditions. The expanded colonies were dissociated using 0.05% trypsin-EDTA solution for passaging.

### MERVL-targeting ASO design

Three independent 20-nt ASOs were designed based on the consensus sequence of MERVL (Supplementary Table 8). ASO candidates that effectively target the internal retroviral regions of MERVL were selected using ChIRP Probe Designer (https://www.biosearchtech.com/support/tools/design-software/chirp-probe-designer), taking into account the secondary structure of the MERVL RNA. To identify potential binding sites for MERVL-targeting ASOs in silico, nonredundant high-confidence MERVL loci were downloaded from RepeatMasker database (http://www.repeatmasker.org/species/mm.html) and converted to BED format. Consequently, FASTA files for these MERVL loci were extracted using the getfasta function as implemented in BEDTools (version 2.30.0)[48] and entered into the makeblastdb program from BLASTn (version 2.6.0+) to generate DNA database for BLASTn search. Each ASO sequence was used to search the database of MERVL DNA using BLASTn[49], with up to two mismatches allowed. The synthesized ASOs were chemically modified with phosphorothioate linkage and 2′-O-methoxyethyl modification at positions 1–5 and 15–20. As the control for the ASO experiments, we also generated a 20-nt randomized oligonucleotide that does not target the mouse genome (as determined by BLASTn search) and three independent 20-nt SOs that were complementary to respective MERVL-targeting ASOs. The three independent ASOs/SOs against MERVL with the highest targeting rate and randomized non-targeting ASO (scrambled ASO) were used for KD experiments (Supplementary Table 8).

### ASO-mediated KD of MERVL in ESCs

A total of $1 × 10^5$ ESC[Dux]s were harvested from culture dishes per one nucleofection. To validate efficiency of the ASO-mediated KD, the cells were nucleofected with 1 nmol ASO against MERVL or scrambled ASO using P3 Primary Cell 96-well Nucleofector Kit (Lonza) according to manufacturer's instruction and seeded into each well of a 24-well plate containing 500 μl ESC medium with 2i, LIF and 1.1 μl iMatrix-511 Silk solution (0.5 mg ml$^{-1}$, Nippi). After 6 h of nucleofection, we added 0.5 μl of 10 μg ml$^{-1}$ Dox to each well of 24-well plate. The following day (24 h after nucleofection), the cells were harvested for quantitative RT-PCR and western blotting to evaluate the expression of MERVL.

### Embryo culture and microinjection

Female mice were superovulated by intraperitoneal administrations of 150 μl CARD HyperOva (Kudo) followed 48 h later with 7.5 IU human chorionic gonadotropin (Asuka Pharmaceutical). After injection of

human chorionic gonadotropin, females were housed with male mice overnight for copulation. The following day, zygotes were collected from the oviducts of superovulated/mated females. The microinjection was carried out under a phase-contrast inverted microscope (IX73, Olympus) equipped with a micromanipulation system (Narishige). Each ASO (20 μM), SO (20 μM) and siRNA (20 and 80 μM) was microinjected into the male pronuclei of zygotes using FemtoJet 4i (Eppendorf). All ASO, SO and siRNA sequences used in this study are listed in Supplementary Table 8. Embryos were cultured in KSOM (MR-101-D, Sigma) at 37°C with 5% $CO_2$.

### RNA rescue assay

ASO-resistant full-length MERVL was constructed by substituting all of the three ASO-targeted sequences of MERVL using inverse PCR. The primers used to produce the constructs are listed in Supplementary Table 8. Briefly, intact MERVL sequence was obtained from a bacterial artificial chromosome plasmid (RP23-231A8, https://www.ncbi.nlm.nih.gov/nuccore/AC127321.4, Thermo Fisher Scientific) and subcloned into pCAGGS_Myc plasmid using NEBuilder HiFi DNA Assembly Master Mix (New England Biolabs). Individual ASO-targeted sequences were mutated by inverse PCR with PrimeSTAR GXL DNA polymerase (TaKaRa) and specific primer sets (Supplementary Table 8). For subsequent steps, ASO-resistant full-length MERVL was amplified by PCR with a specific primer set containing the T7 promoter sequence at the 5′ end of forward primer for in vitro transcription with T7 polymerase. Following PCR amplification, in vitro transcription was performed using MEGAscript T7 Transcription Kit (Thermo Fisher Scientific) according to the manufacturer's instructions. Transcribed RNA was purified using RNeasy Mini Kit (QIAGEN). ASO-resistant MERVL RNA was co-microinjected into the male pronucleus of zygotes with 20 μM ASOs (mixed) at the final concentration of 200 ng μl$^{-1}$.

### CasRx-mediated KD of MERVL

The expression plasmids of CasRx (PB-CAG-CasRx-P2A-EGFP-BGH PolyA, #154004, Addgene) and gRNA (pXR003: CasRx gRNA cloning backbone, #109053, Addgene) were gifted from H. Yang and P. Hsu[22,50]. For optimal expression of CasRx in early stages of preimplantation embryos, the CAG promoter sequence in the CasRx-expressing plasmid was replaced with CBh promoter, which was obtained from pX330-U6-Chimeric_BB-CBh-hSpCas9 (pX330, gifted from F. Zhang, #42230, Addgene)[51], using NEBuilder HiFi DNA Assembly Cloning Kit (New England Biolabs). The pXR003 CasRx gRNA-expressing plasmid was linearized by FastDigest BpiI (Thermo Fisher Scientific), an isoschizomers of BbsI, and then individual annealed gRNA spacer with four overhangs (sense: 5′-AAAC, antisense: 5′-AAAA) was inserted into the linearized pXR003 CasRx gRNA-expressing plasmid by T4 DNA ligase (Ligation high v2, TOYOBO). The primers and gRNA sequences used to produce the constructs are listed in Supplementary Table 8. The expression plasmids of CasRx and MERVL-targeting gRNA (or non-targeting gRNA) were microinjected into the male pronucleus of zygotes at a final concentration of 10 ng μl$^{-1}$ each.

### CRISPRi targeting MERVL

For evaluation of *cis*-acting functions of MERVL, we used the CRISPRi system, composed of dCas9-KRAB-MeCP2, which specifically induces H3K9 trimethylation at target loci. We designed three independent gRNAs targeting the coding regions of Gag and Pol in full-length copies of MERVL to repress MERVL transcription in preimplantation embryos. The CBh-dCas9-KRAB-MeCP2 fusion construct was constructed using the pX330 (#42230, Addgene)[51] plasmid backbone. dCas9-KRAB-MeCP2 sequence was obtained from a PB-TRE-dCas9-KRAB-MeCP2 (gifted from A. Califano, #122267, Addgene) plasmid and subcloned into pX330 using NEBuilder HiFi DNA Assembly Cloning Kit (New England Biolabs). Constructed pX330 plasmid in which hSpCas9 site had been replaced with dCas9-KRAB-MeCP2 (hereafter referred as pX330-CRISPRi), was

then subjected to subcloning of sgRNA sequences. Individual annealed sgRNA spacer with four overhangs (sense: 5′-CACC, antisense: 5′-AAAC) was subcloned into the BbsI site of the pX330 plasmid[51]. Subsequently, a transcriptional unit of each gRNA vector (consisting of a human U6 promoter, gRNA spacer and scaffold sequences) was inserted into SacII-to-XbaI sites of pX330-CRISPRi plasmid. The primers and gRNA sequences used to produce the constructs are listed in Supplementary Table 8. MERVL-targeting pX330-CRISPRi plasmid was microinjected into the male pronucleus of zygotes at a final concentration of 15 ng μl⁻¹.

## smFISH and immunofluorescence analysis

Embryos were fixed using fixative solution (4% paraformaldehyde and 0.2% polyvinyl alcohol (PVA) in PBS) for 20 min at room temperature (RT) and then permeabilized with 0.5% TritonX-100 in PBS for 10 min at RT. For smFISH, embryos were washed once with Stellaris Wash Buffer A (Bioresearch Technologies) for 5 min at RT. FISH was then performed overnight at 37°C in Stellaris Hybridization Buffer (Bioresearch Technologies) containing 25 pmol of a 48-probe set (Quasar 670-labeled, Bioresearch Technologies) against MERVL RNA. The following day, embryos were washed once with Stellaris Wash Buffer A for 30 min at 37°C and then counterstained with DAPI (1 μg ml⁻¹, Nacalai Tesque) for 30 min at 37°C. After washing with Stellaris Wash Buffer B (Bioresearch Technologies) for 5 min at RT, embryos were transferred to 10 μl drops of 0.2% PVA in PBS on a glass-bottomed dish covered by paraffin oil. The FISH probe sequences used in this study are listed in Supplementary Table 8. For immunofluorescence analysis, nonspecific immunoreaction was blocked by incubating embryos in 2% bovine serum albumin (BSA) in PBS for 1 h at RT. After blocking, whole-mount immunofluorescence analyses were conducted using the following primary antibodies: mouse anti-MERVL-Gag (1/5 dilution, kept in our lab), rabbit anti-OCT4 (1/100 dilution, ab181557, abcam), mouse anti-CDX2 (1/100 dilution, ab157524, abcam), mouse anti-E-Cadherin (1/100 dilution, 610182, BD Transduction Laboratories), rabbit anti-cleaved caspase-3 (1/200 dilution, 9661, CST) and rabbit anti-Phospho-S15-p53 (1/200 dilution, 9284, CST). Embryos were incubated overnight at 4°C with primary antibody. The following day, washed embryos three times with 0.5% BSA in PBS for 5 min each and then incubated with Alexa Fluor 488- and/ or 568-conjugated anti-mouse and/or anti-rabbit immunoglobulin G secondary antibodies (1/500 dilution, Thermo Fisher Scientific) for 1 hr at RT. After washing three times with 0.5% BSA in PBS for 10 min each, the embryos were counterstained with DAPI (1 μg ml⁻¹) for 30 min at RT and transferred to 10 μl drops of 0.2% PVA in PBS on a glass-bottomed dish covered by paraffin oil. smFISH and immunofluorescence images were obtained with a BZ-X810 fluorescence microscope (Keyence) or FV3000 confocal laser scanning microscope (Olympus) and processed with ImageJ (NIH)[52].

## EU incorporation assay

The embryos microinjected with either scramble ASOs or MERVL ASOs, were cultured in KSOM for 24 hrs. On the next day, two-cell-stage embryos were transferred to new KSOM containing 1 mM EU (Click-iT RNA Alexa Fluor 488 Imaging Kit, Thermo Fisher Scientific) and cultured for 1 h at 37°C with 5% CO₂. The EU-labeled embryos were immediately fixed and permeabilized as described above. Incorporated EU into newly synthesized RNA was detected using the Click-iT RNA Alexa Fluor 488 Imaging Kit (Thermo Fisher Scientific) according to the manufacturer's instructions. In brief, the fixed embryos were treated with the Click-iT reaction cocktail containing Alexa Fluor 488 azide for 30 min at RT, and then washed once with Click-iT reaction rinse buffer. After washing once with 0.2% PVA in PBS, the embryos were counterstained with Hoechst 33342 (2 μg ml⁻¹, Dojindo) for 30 min at RT and transferred to 10 μl drops of 0.2% PVA in PBS on a glass-bottomed dish covered by paraffin oil. Images of EU-labeled embryos were obtained with a FV3000 confocal laser scanning microscope (Olympus) and processed with ImageJ (NIH)[52].

## Quantitative RT-PCR

Acidic Tyrode's solution (Sigma) containing 0.2% PVA was used to remove the zona pellucida (ZP) from embryos before sample collection. Then, 30 ZP-free morula-stage embryos and nucleofected ESC^Dux's were lysed in Lysis Solution (from SuperPrep II Cell Lysis & RT Kit, TOYOBO), directly. Genomic DNA elimination and reverse transcription were performed using the SuperPrep II Cell Lysis & RT Kit with random hexamers and oligo dT primers according to the manufacturer's instructions. Real-time quantitative PCR was carried out using the following conditions: 95°C for 1 min, followed by 45 cycles each of 95°C for 15 s and 60°C for 1 min, on a Thermal Cycler Dice Real Time System III (TaKaRa) with Thunderbird SYBR qPCR Mix (TOYOBO) and specific primer sets (Supplementary Table 8). Relative gene expression was quantified with the ΔΔCT method and normalized to *Actb* or *Gapdh* expression.

## Western blotting

The cells were directly lysed in Laemmli SDS sample buffer (1×, 62.5 mM Tris-HCl (pH 6.8), 2% SDS, 10% glycerol, 5% β-mercaptoethanol and 0.02% bromophenol blue) and sonicate with Bioruptor at a high setting, for 10 cycles each of 30 s with 30 s intervals. Total proteins were denatured at 95°C for 5 min and separated by 10% SDS-PAGE. The proteins were then transferred onto a Protran Nitrocellulose Membranes (0.45 μm pore size, GE Healthcare) via Power Blotter-Semi-dry Transfer System (Thermo Fisher Scientific). The membrane was blocked with Bullet Blocking One for Western Blotting (Nacalai Tesque) for 10 min at RT with gently rocking before incubation with mouse anti-MERVL-Gag (1/5 dilution, kept in our lab) or mouse anti-β-tubulin (1/2,000 dilution, E7, Developmental Studies Hybridoma Bank) antibody in 1:20 diluted blocking buffer with 0.1% PBST at 4°C overnight. The following day, the membranes were then incubated with HRP-conjugated goat anti-mouse immunoglobulin G secondary antibody (1/10,000 dilution, #330, MBL Life Science) in 1:20 diluted blocking buffer with 0.1% PBST for 30 min at RT with gently rocking. After washing three times with 0.1% PBST for 10 min each, Blots were developed using ECL Western Blotting Detection Reagent (Sigma) and exposed onto X-ray film.

## RNA-seq

Preparation of total RNA-seq libraries was performed using SMART-seq Stranded Kit (Clontech), according to the manufacturer's instructions. In brief, 30 ZP-free two-cell, four-cell and eight-cell stage embryos whose cleavage stages were visually confirmed under the microscope, were lysed in 1× Lysis Buffer containing RNase inhibitor (0.2 IU μl⁻¹, from SMART-seq Stranded Kit, Clontech), directly. RNAs were randomly sheared by heating at 85°C for 8 min and subjected to reverse transcription with random hexamers and PCR amplification. Ribosomal fragments were depleted from each cDNA sample with scZapR and scR-Probes. Indexed total RNA-seq libraries were enriched through a second PCR amplification and sequenced using an Illumina HiSeqX sequencer (paired end, 150 bp). Two biological replicates were generated for each sample.

## miniATAC-seq

The protocol for miniATAC-seq library was adapted from a previous report with minor modifications[36]. Briefly, 30 ZP-free two-cell, four-cell and eight-cell stage embryos whose cleavage stages were visually confirmed under the microscope were lysed in 6 μl lysis buffer (10 mM Tris-HCl (pH 7.4), 10 mM NaCl, 3 mM MgCl₂ and 0.5% NP-40) for 10 min on ice. After cell lysis, 2 μl ddH₂O, 10 μl of 2× TD Buffer (from Nextera XT DNA Prep Kit, Illumina) and 2 μl of Tn5 Transposase (from Nextera XT DNA Prep Kit, Illumina) were added to the lysates, which were mixed by pipetting followed by incubation at 37°C for 30 min. To stop the transposase reaction, 5 μl of 0.5% SDS was added and incubated at RT for 5 min. After Tn5 tagmentation, 100 ng Carrier RNA (Yeast tRNA, Sigma) was added and then messed samples up to 130 μl with 1× TE Buffer (10 mM Tris-HCl (pH 8.0) and 1 mM EDTA). Tagmented DNA was

purified by phenol-chloroform extraction and ethanol precipitation with 20 μg glycogen as carrier and dissolved in 20 μl ddH$_2$O. Afterwards, PCR was performed to amplify and index libraries using the following conditions: 72 °C for 5 min and 98 °C for 30 s, followed by 18 cycles each of 98 °C for 10 s, 63 °C for 30 s and 72 °C for 1 min, with NEBNext HiFi PCR Master Mix and specific index primer sets (Supplementary Table 8). Pooled miniATAC-seq libraries were sequenced using an Illumina HiSeqX sequencer (paired end, 150 bp). Two biological replicates were generated for each sample.

### RNA-seq and ATAC-seq data processing

Methods for RNA-seq analysis have been described previously[53]. In short, raw paired-end RNA-seq reads were aligned to indexed mouse genome (GRCm38/mm10) using STAR aligner version 2.5.3a[54] with following options: --twopassMode Basic; --outSAMtype BAM SortedByCoordinate; --outFilterType BySJout; --outFilterMultimapNmax 1; --winAnchorMultimapNmax 50; --chimSegmentMin 12; --chimJunctionOverhangMin 8; --alignSJoverhangMin 8; --alignSJDBoverhangMin 10; --outFilterMismatchNmax 999; --outFilterMismatchNoverReadLmax 0.04; --alignSJstitchMismatchNmax 5 -1 5 5; --outSAMattrRGline ID:GRPundef; --alignIntronMin 20; --alignIntronMax 1000000; --alignMatesGapMax 1000000 for unique alignments. To quantify aligned reads on annotated genes (GENCODE vM25) and repetitive loci (originating from mm10.fa.out (RepeatMasker), best-match TE annotation defined previously[53], we used the featureCounts function, which is part of the Subread package[55]. To detect differentially expressed genes and repetitive elements between control and MERVL-KD embryos, an output file of featureCounts was entered into the DESeq2 package (version 1.16.1)[56]; then, the program functions DESeqDataSetFromMatrix and DESeq were used to compare each gene's expression level between two biological samples. Differentially expressed genes were identified through two criteria: (1) at least twofold change and (2) binominal tests ($P$adj < 0.05; $P$ values were adjusted for multiple testing using the Benjamini–Hochberg method). To perform GO and ChEA analyses, we used the functional annotation clustering tool in Enrichr[57]. The annotated terms with $P$ < 0.05 from a modified Fisher's exact test were considered significant. To visualize read enrichments over representative genomic loci, TDF files were created from sorted BAM files using the IGVTools count function (Broad Institute)[58]. Figures of continuous tag counts over selected genomic intervals were created in the IGV browser (Broad Institute)[58]. To comprehensively evaluate the expression profiles across the entire transcribed regions (genic and intergenic region) in the genome, alignment file from STAR was entered into the groHMM package (version 1.24.0)[41]; then, program function detectTranscripts with default parameter settings was used for detection of TARs de novo based on a two-state hidden Markov model. TARs were further compartmentalized into aTARs and uTARs according to overlap with existing known annotations (in GENCODE vM25 and RepeatMasker) using subtract function from BEDTools (version 2.30.0)[48]. After quantification of aligned reads on uTARs, differentially expressed uTARs between control and MERVL-KD embryos were detected using DESeq2 package[56] with the threshold of $P$adj < 0.01.

Raw paired-end ATAC-seq reads were filtered by TrimGalore (version 0.6.4, https://github.com/FelixKrueger/TrimGalore) with the default setting and aligned to indexed mouse genome (GRCm38/mm10) using bowtie2 (version 2.4.4)[59] with the following options, -N 1; -L 25; --no-mixed; --no-discordant. Multiple aligned reads and PCR duplicates were removed using grep -v 'XS:' and MarkDuplicates function with REMOVE_DUPLICATES = true option, a part of Picard tools. Using SeqMonk (Babraham Bioinformatics), we calculated Pearson correlation coefficients between biological replicates. Peak calling for ATAC-seq data was performed using MACS2 (version 2.1.4)[60] with default arguments; we used a cut-off of $P \leq 10^{-2}$. The average tag density plots were drawn using Ngsplot (version 2.47.1) and plotHeatmap

program as implemented in deepTools (version 3.1.3)[61]. To visualize read enrichment over representative genomic loci, TDF files were created from sorted BAM files using the IGVTools count function (Broad Institute)[58]. Figures for continuous tag counts over selected genomic intervals were created in the IGV browser (Broad Institute)[58]. For $k$-means clustering of ATAC-seq peaks, we firstly generated a set of regions that were called peaks with MACS2 in at least one of the samples, by merging ATAC-seq peak regions of all biological replicates using the mergeBed function from BEDTools (version 2.30.0)[48]. Using this merged peak file and ATAC-seq data from control embryos, we ran $k$-means clustering analysis using computeMatrix and plotHeatmap program as implemented in deepTools (version 3.1.3)[61] and determined that $k$ = 5 was suitable for our data. To evaluate the functional annotation of each cluster, we used Genomic Region Enrichment of Annotation Tool (GREAT, version 4.0.4)[62] and HOMER (version 4.9)[63], which associates ATAC-seq peaks in each cluster with their genomic feature and ontology of their putative target genes adjacent to peaks.

### Statics and reproducibility

All statistical methods, sample sizes and $P$ values for each plot are listed in the figure legends and/or in the corresponding Methods section. In brief, all grouped data are represented as mean ± s.e.m. All box plots are represented as follows: center lines, median; box edges, interquartile range (25 and 75 percentiles); whisker, 90% of the data points. Statistical significance for pairwise comparisons were determined using the two-sided unpaired $t$-tests, chi-square tests and Mann–Whitney $U$-tests with Bonferroni correction. All quantitative data are represented from three or more biological replicates. Fisher's exact test and hypergeometric test were used for the detection of significantly enriched GO terms, genes, and loci compared with backgrounds. Differentially expressed genes, TEs and loci were determined in the DESeq2 package[56]. Next-generation sequencing data (total RNA-seq and miniATAC-seq) are based on two biological replicates. For all experiments, no statistical methods were used to predetermine sample size. Experiments were not randomized, and investigators were not blinded to allocation during experiments and outcome assessments.

### Reporting summary

Further information on research design is available in the Nature Portfolio Reporting Summary linked to this article.

## Data availability

The total RNA-seq and miniATAC-seq data in embryos upon MERVL-KD, MERVLi and their respective controls are deposited in the Gene Expression Omnibus under accession code GSE196520. Source data are provided with this paper.

## Code availability

This study did not make any custom code or algorithm. All software used in this study is publicly available and cited in the main text and/or Methods section.

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

## Acknowledgements

We thank all members of the Siomi laboratory for discussions on this work, K. Hayashi (Graduate School of Medicine, Faculty of Medicine, Osaka University) and A. Inoue (Center for Integrative Medical Science, RIKEN) for critical reading of the manuscript, and S. Aikawa (Graduate School of Medicine, The University of Tokyo) and C. Takeuchi (Keio University School of Medicine) for assistance with NGS data analyses. This work was supported by JSPS Grants-in-Aid for Early-Career Scientists (21K15108 to A.S.), the Kato Memorial Bioscience Foundation Research Grant (to A.S.), JST Fusion Oriented Research for disruptive Science and Technology (JPMJFR214O to A.S.), MEXT Grants-in-Aid for Scientific Research in Innovative Areas (19H05753 to H.S. and 22H04700 to A.S.), AMED project for elucidating and controlling mechanisms of aging and longevity (1005442 to H.S.), the Uehara Memorial Foundation Research Incentive Grant (to A.S.) and the Uehara Memorial Foundation Research Grant (to H.S.).

## Author contributions

The manuscript was written by A.S., T.K. and H.S., with critical feedback from all other authors. A.S., T.K., H.I. and H.S. conceived and designed this study. K.M. provided instruction for biochemistry and molecular biology. H.I. developed ASOs against MERVL. T.K. and A.S. performed microinjection for generating MERVL-KD embryos and RNA rescue assay with the help of H.M. and carried out phenotypic analyses of generated MERVL-KD embryos. A.S. and T.K. performed total RNA-seq and ATAC-seq analyses. T.K., Y.G. and A.S. designed and interpreted bioinformatic analyses with the help of M.A. A.S. and H.S. supervised the project.

## Competing interests

The authors declare no competing interests.

## Additional information

**Extended data** is available for this paper at https://doi.org/10.1038/s41588-023-01324-y.

**Correspondence and requests for materials** should be addressed to Haruhiko Siomi.

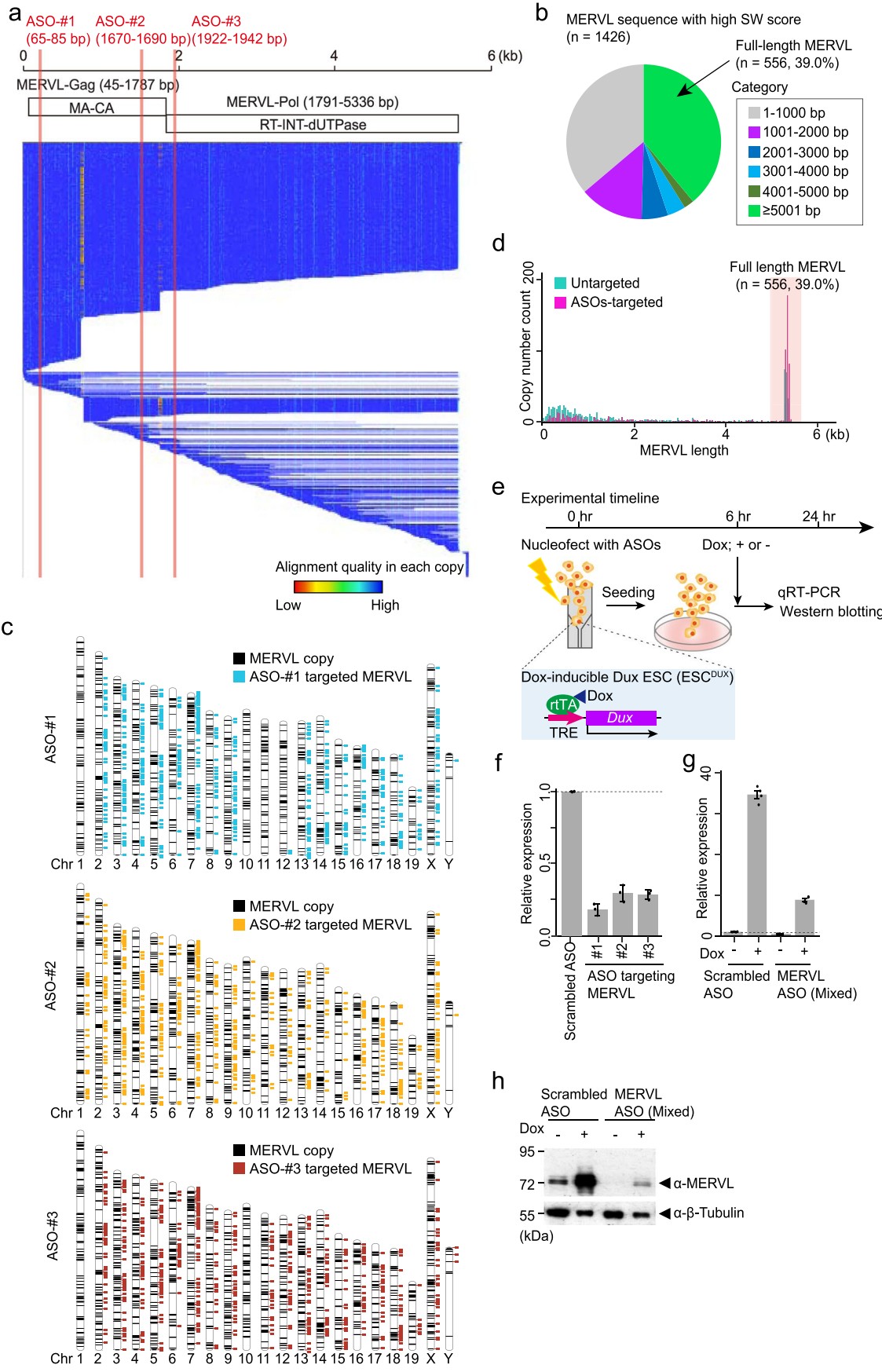

**Extended Data Fig. 1 | See next page for caption.**

**Extended Data Fig. 1 | Design and validation of ASOs targeting MERVL. (a)** Schematic of MERVL indicating the positions of ASOs. Intact MERVL encodes the retroviral proteins; Gag (Group-specific antigens, comprised of MA, matrix; CA, capsid proteins) and Pol (Polymerase, comprised of RT, reverse transcriptase; INT, integrase; dUTPase, dUTP phosphatase). Below whisker plot showing alignment quality in each interspersed genomic MERVL copy (adapted from Dfam: https://dfam.org/family/DF0003918/seed). **(b)** Pie chart indicates the populations of full length (≥ 5001 bp) and truncated (1–5000 bp) MERVL copies across the mouse genome. **(c)** Chromosome maps showing the distribution of MERVL copies that are targeted by ASOs throughout the mouse genome. Each colored rectangle (blue for ASO-#1, yellow for ASO-#2, and red for ASO-#3) indicates targeted MERVL copy. **(d)** Histogram showing the distributions of ASO-targeted (magenta) and untargeted (green) MERVL copies. The abundance of each length of MERVL element was tallied with a given count. The proportion of full-length MERVL (>5 kb) was highlighted in red. **(e)** Schematic for ASO-mediated KD against MERVL in ESC harboring a doxycycline (Dox)-inducible Dux transgene (termed ESC$^{DUX}$). TRE, tetracycline responsive element; rtTA, reverse tetracycline responsive transcriptional activator. **(f)** Expression levels of MERVL mRNA measured by qRT-PCR in ESC$^{DUX}$s nucleofected with each individual MERVL-targeting ASO and scrambled ASO. Relative expression is quantified with ΔΔCt method and normalized to *Gapdh* expression. Bars show means with s.e.m. Dots represent biological replicates (n = 3 samples). **(g)** Expression level of MERVL mRNA measured by qRT-PCR in scrambled control and ASOs (Mixed)-mediated MERVL-KD ESC$^{DUX}$s in the presence or absence of Dox. Relative expression is quantified with ΔΔCt method and normalized to *Gapdh* expression. Bars show means with s.e.m. Dots represent biological replicates (n = 4 samples). **(h)** Expressions of MERVL retroviral protein (Gag), measured by western blotting in scrambled control and ASOs (Mixed)-mediated MERVL-KD ESC$^{DUX}$s in the presence or absence of Dox. β-Tubulin was used as a loading control. Representative blot image is from 3 independent experiments. Data for panels in b-d and f-h are available as source data.

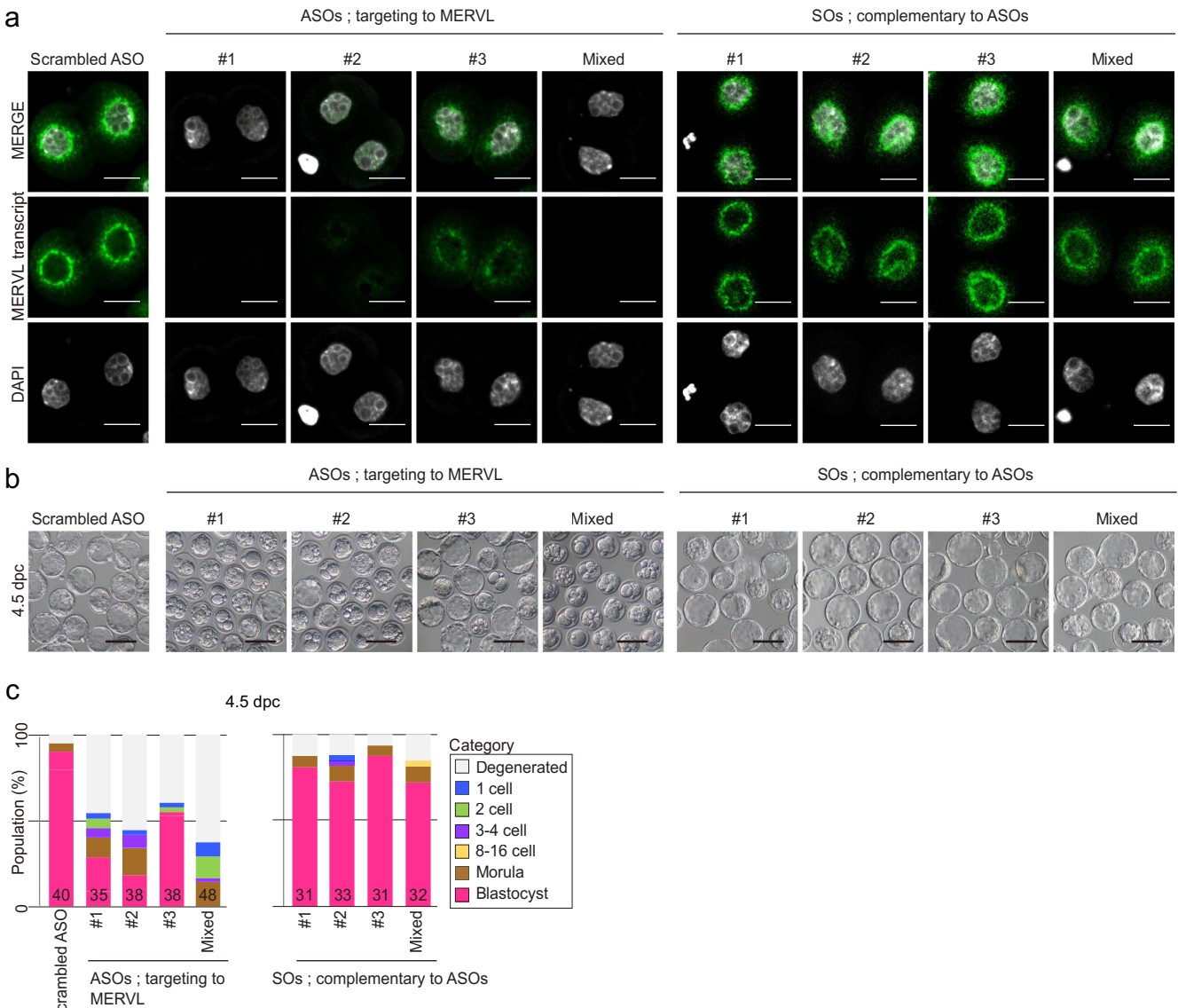

**Extended Data Fig. 2 | The KD effectiveness of each ASO against MERVL in preimplantation embryos.** (**a**) Representative images of smFISH for MERVL RNA (green) with DAPI counterstain (gray) in two-cell stage embryos from each experimental condition, from 3 independent experiments. Zygotes were injected with scrambled ASO, MERVL ASOs or the corresponding SOs. Scale bar: 20 μm.

(**b**) Representative phase-contrast images of 4.5 dpc blastocysts from each experimental condition, from 2 independent experiments. Scale bar: 100 μm. (**c**) Percentage of embryos by developmental stage at 4.5 dpc in each experimental condition. The number of embryos in each experimental condition is shown in the bottom. Data for panels in c are available as source data.

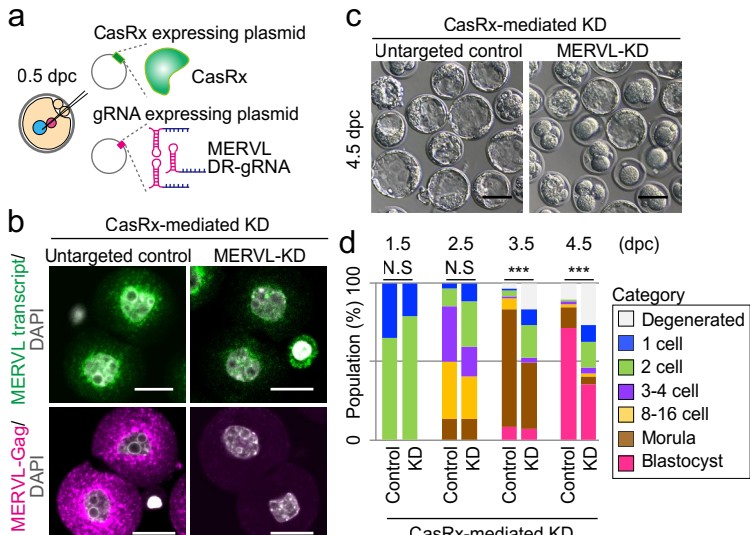

**Extended Data Fig. 3 | CasRx-mediated MERVL-KD and its effects on preimplantation development.** (**a**) Schematic of experimental procedure for CasRx-mediated KD of MERVL. (**b**) Representative images of smFISH and immunofluorescence staining for MERVL RNA (top) and MERVL-Gag protein (bottom) with DAPI counterstain in untargeted control and MERVL-KD two-cell stage embryos at 1.5 dpc, from 3 independent experiments. Scale bar: 20 µm. (**c**) Representative phase-contrast images of 4.5 dpc blastocysts upon untargeted control and MERVL-KD conditions, from 3 independent experiments. Scale bar: 100 µm. (**d**) Percentage of embryos by stages of development, upon untargeted control (n = 69) and MERVL-KD (n = 70) conditions. N.S., not significant; ***$P < 0.001$, chi-square test. Data for panel in d is available as source data.

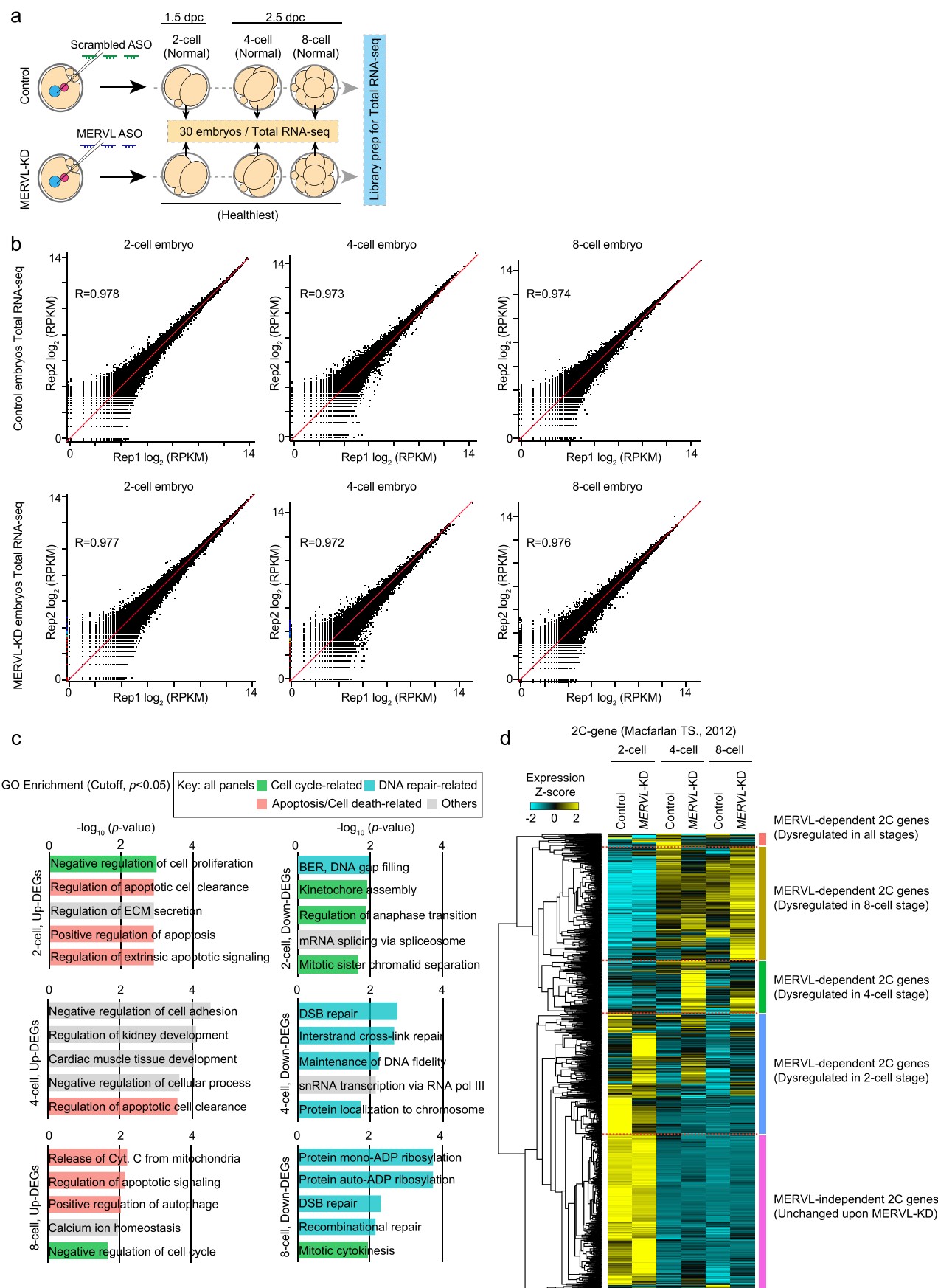

**Extended Data Fig. 4 | See next page for caption.**

**Extended Data Fig. 4 | Total RNA-seq analysis of control and MERVL-KD embryos at two-cell, four-cell, and eight-cell stages.** (**a**) Schematic of sample collection for total RNA-seq analysis in scrambled control and ASO-mediated MERVL-KD embryos. We supplied the apparently healthiest embryos upon MERVL-KD for preparation of a total RNA-seq library. (**b**) Scatter plots showing the reproducibility between biological replicates in total RNA-seq data. Pearson correlation values (R) are shown. (**c**) GO analysis of DEGs in MERVL-KD embryos, assessed by the Enrichr. (**d**) Heatmap showing hierarchical clustering of expression patterns of all 2C genes (as defined in 12) in control and MERVL-KD embryos at two-cell, four-cell and eight-cell stages. Expression level of each gene is shown in Z-score, calculated by subtracting the mean expression value and dividing by standard deviation. Data for panels in c and d are available as source data.

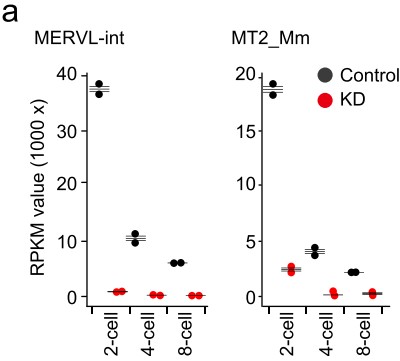

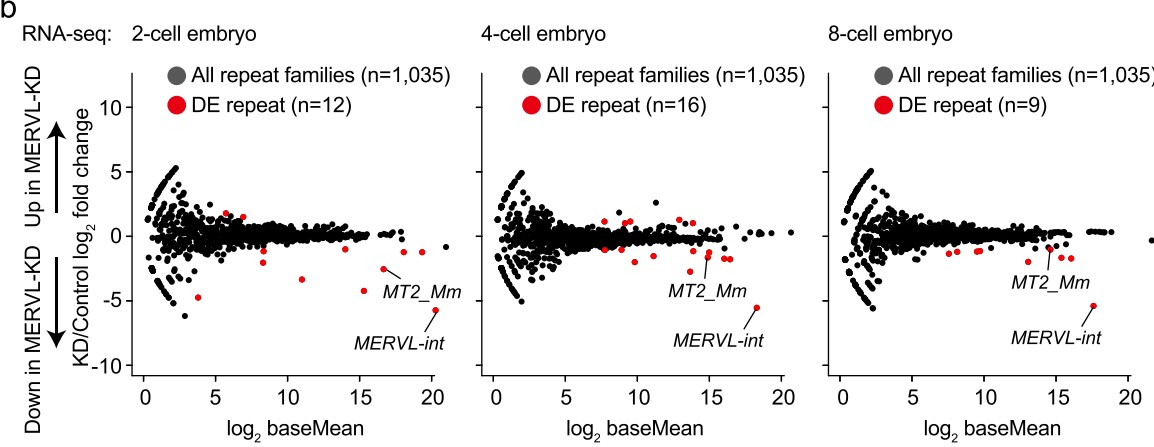

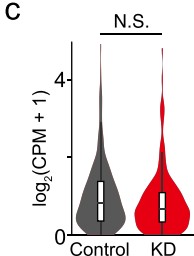

**Extended Data Fig. 5 | MERVL-KD only dysregulates a few types of transposable element.** (**a**) Dot plots showing RPKM values for MERVL-int and MT2_Mm in scrambled control and MERVL-KD embryos at two-cell, four-cell, and eight-cell stages. Central bars represent medians, the top and bottom lines encompass 50% of the data points. (**b**) MA plots show differentially expressed (DE) repetitive elements (annotated in RepeatMasker) between scrambled control and MERVL-KD embryos. DE repeats were defined as those with a |FoldChange| ≥ 2 and $P$adj < 0.05 (binomial test with Benjamini–Hochberg correction) and shown in red circles. MERVL-int and MT2_Mm were included in downregulated repeats. (**c**) Violin plots indicate CPM values for chimeric fusion transcripts in scrambled control and MERVL-KD two-cell stage embryos. Each plot encompasses box plot; central bars represent medians, box edges indicate 50% of data points, and the whiskers show 90% of data points. N.S., not significant, two-tailed unpaired t-tests. Data for panels in a-c are available as source data.

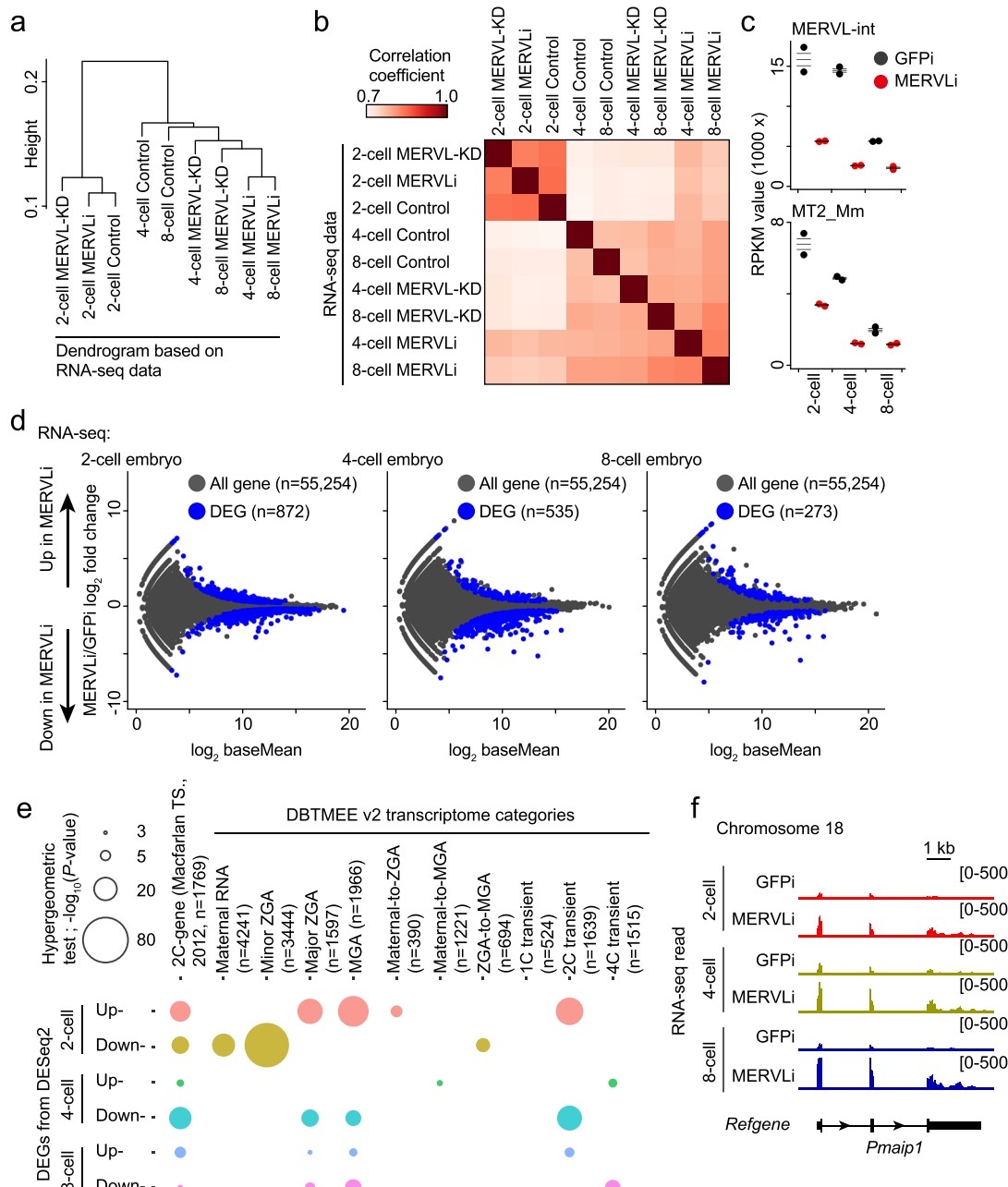

**Extended Data Fig. 6 | RNA-seq analysis upon CRISPRi induction shows that MERVLi embryos resemble ASO-mediated MERVL-KD embryos. (a)** Unsupervised hierarchical cluster of all annotated transcript profiles (n = 55,254) from RNA-seq data in embryos from different groups and developmental stages. Each dendrogram leaf represents an RNA-seq sample and the y-axis shows the distance based on Pearson correlation between each pair of samples. **(b)** Heatmap showing pairwise Pearson correlation coefficient of gene expression between each pair of samples. **(c)** Dot plots showing RPKM values for MERVL-int and MT2_Mm in GFPi control and MERVLi embryos. Central bars represent medians, the top and bottom lines encompass 50% of the data points. **(d)** RNA-

seq differential gene expression analysis: MERVLi versus GFPi control embryos obtained at two-cell, four-cell and eight-cell stages. 872, 535 and 273 genes evinced significant changes in expression in MERVLi embryos (blue circles, $P$adj < 0.05; binomial test with Benjamini–Hochberg correction). **(e)** Bubble plot showing overlap between all DEGs in MERVLi embryos with the list of 2C genes and DBTMEE v2 transcriptome categories. The bubble plot sizes show the -$\log_{10}[P$ values] derived from a hypergeometric test. **(f)** Track views show RNA-seq signals in GFPi control and MERVLi embryos, on a representative 2C gene locus (as defined in 12). The y-axis represents normalized tag counts for total RNA-seq in each sample. Data for panels in a-e are available as source data.

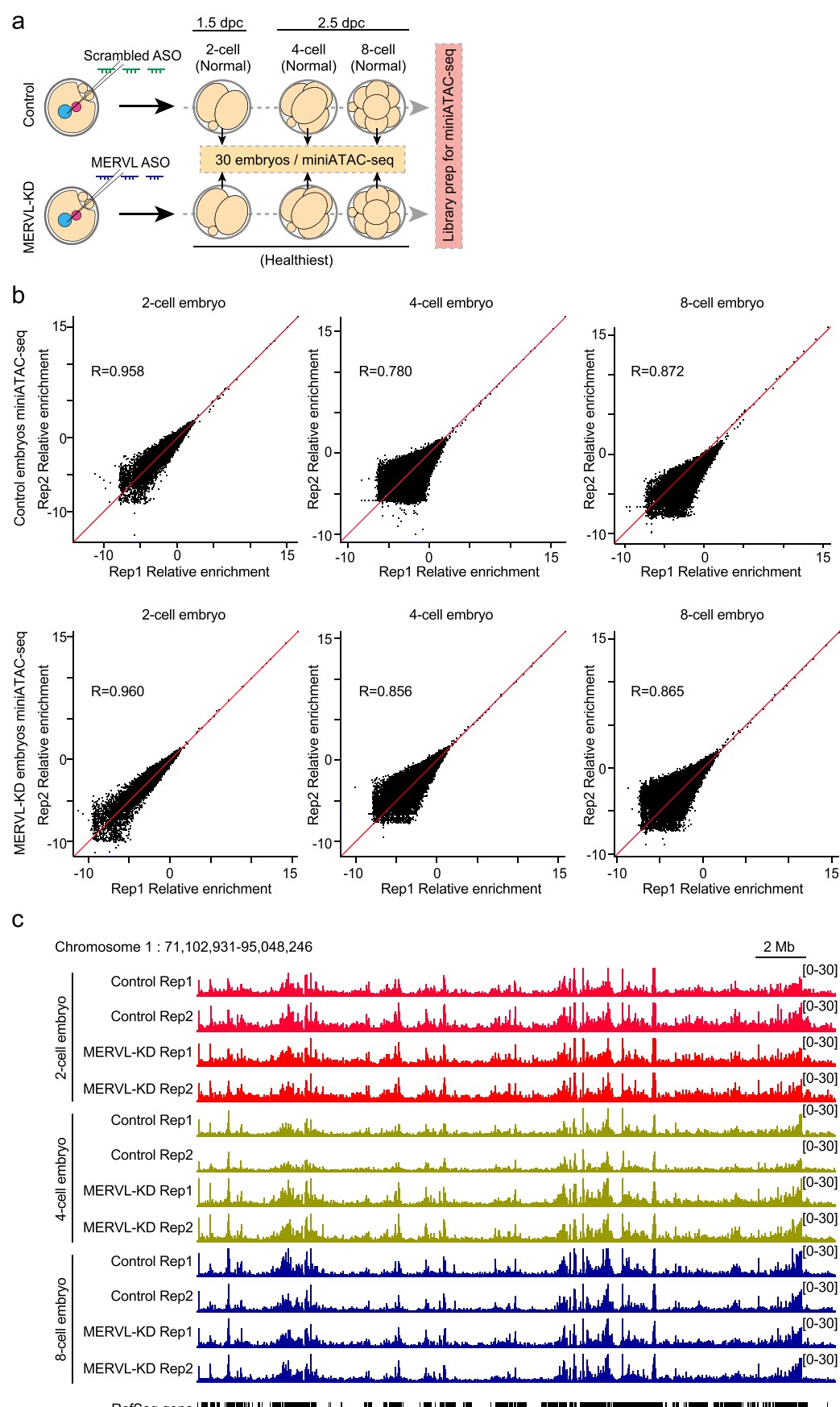

**Extended Data Fig. 7 | See next page for caption.**

**Extended Data Fig. 7 | miniATAC-seq analysis of control and MERVL-KD embryos at two-cell, four-cell, and eight-cell stages.** (**a**) Schematic of sample collection for miniATAC-seq analysis in scrambled control and ASO-mediated MERVL-KD embryos. We supplied apparently healthiest embryos upon MERVL-KD for preparation of miniATAC-seq library. (**b**) Scatter plots showing the reproducibility between biological replicates in miniATAC-seq data. Pearson correlation values (R) are shown. (**c**) Track view of ATAC-seq read enrichments at a representative chromosomal position in all biological replicates. The y-axis represents normalized tag counts for ATAC-seq in each sample.

# Reporting Summary

## Statistics

For all statistical analyses, confirm that the following items are present in the figure legend, table legend, main text, or Methods section.

| n/a | Confirmed | |
|---|---|---|
| ☐ | ☒ | The exact sample size (*n*) for each experimental group/condition, given as a discrete number and unit of measurement |
| ☐ | ☒ | A statement on whether measurements were taken from distinct samples or whether the same sample was measured repeatedly |
| ☐ | ☒ | The statistical test(s) used AND whether they are one- or two-sided<br>*Only common tests should be described solely by name; describe more complex techniques in the Methods section.* |
| ☒ | ☐ | A description of all covariates tested |
| ☐ | ☒ | A description of any assumptions or corrections, such as tests of normality and adjustment for multiple comparisons |
| ☐ | ☒ | A full description of the statistical parameters including central tendency (e.g. means) or other basic estimates (e.g. regression coefficient) AND variation (e.g. standard deviation) or associated estimates of uncertainty (e.g. confidence intervals) |
| ☐ | ☒ | For null hypothesis testing, the test statistic (e.g. *F*, *t*, *r*) with confidence intervals, effect sizes, degrees of freedom and *P* value noted<br>*Give P values as exact values whenever suitable.* |
| ☒ | ☐ | For Bayesian analysis, information on the choice of priors and Markov chain Monte Carlo settings |
| ☒ | ☐ | For hierarchical and complex designs, identification of the appropriate level for tests and full reporting of outcomes |
| ☐ | ☒ | Estimates of effect sizes (e.g. Cohen's *d*, Pearson's *r*), indicating how they were calculated |

*Our web collection on statistics for biologists contains articles on many of the points above.*

## Software and code

Policy information about availability of computer code

| Data collection | Data were collected with GNU Wget, 64-bit miniconda3 package 'conda install' and SRA Toolkit 'prefetch' from the following web databases. |
|---|---|
| Data analysis | Source code for all software and tools used in this study with documentation, examples and additional information, is available at following URLs:<br>https://github.com/GenomeImmunobiology/Sakashita_et_al_2020 (best-match TE annotation set)<br>https://dfam.org/family/DF0003918/summary (MERVL sequence)<br>https://www.biosearchtech.com/support/tools/design-software/stellaris-probe-designer (Stellaris Probe Designer for smFISH)<br>https://bedtools.readthedocs.io/en/latest (bedtools v2.30.0)<br>https://www.ncbi.nlm.nih.gov/books/NBK279690 (BLAST v2.6.0+)<br>https://www.ncbi.nlm.nih.gov/geo/query/acc.cgi?acc=GSE45719 (scRNA-seq dataset in mouse preimplantation embryos)<br>https://github.com/alexdobin/STAR (STAR RNA-seq aligner v2.5.3a)<br>http://hgdownload.cse.ucsc.edu/goldenpath/mm10/bigZips/mm10.fa.gz (GRCm38/mm10)<br>https://ftp.ebi.ac.uk/pub/databases/gencode/Gencode_mouse/release_M25/gencode.vM25.annotation.gtf.gz (GENCODE gene annotation)<br>http://subread.sourceforge.net (Subread v2.0.1)<br>https://bioconductor.org/packages/release/bioc/html/DESeq2.html (DESeq2 v1.16.1)<br>https://maayanlab.cloud/Enrichr (Enrichr)<br>https://dbtmee.hgc.jp (DBTMEE v2, Database of Transcriptome in Mouse Early Embryos)<br>https://software.broadinstitute.org/morpheus (Morpheus)<br>https://software.broadinstitute.org/software/igv/igvtools (IGVTools v2.9.2)<br>https://www.bioinformatics.babraham.ac.uk/projects/seqmonk (SeqMonk v1.48.0)<br>http://bowtie-bio.sourceforge.net/bowtie2 (bowtie2 v2.4.4)<br>https://hbctraining.github.io/Intro-to-ChIPseq/lessons/05_peak_calling_macs.html (MACS2 v2.1.4)<br>https://deeptools.readthedocs.io/en/develop (deepTools v3.1.3) |

For manuscripts utilizing custom algorithms or software that are central to the research but not yet described in published literature, software must be made available to editors and reviewers. We strongly encourage code deposition in a community repository (e.g. GitHub). See the Nature Portfolio guidelines for submitting code & software for further information.

## Data

Policy information about availability of data

All manuscripts must include a data availability statement. This statement should provide the following information, where applicable:

- Accession codes, unique identifiers, or web links for publicly available datasets
- A description of any restrictions on data availability
- For clinical datasets or third party data, please ensure that the statement adheres to our policy

The total RNA-seq and miniATAC-seq data reported in this study are deposited in the Gene Expression Omnibus (GEO) under accession code GSE196520. Other scRNA-seq dataset from GSE45719 are described and cited in the manuscript.

# Field-specific reporting

Please select the one below that is the best fit for your research. If you are not sure, read the appropriate sections before making your selection.

☒ Life sciences   ☐ Behavioural & social sciences   ☐ Ecological, evolutionary & environmental sciences

For a reference copy of the document with all sections, see nature.com/documents/nr-reporting-summary-flat.pdf

# Life sciences study design

All studies must disclose on these points even when the disclosure is negative.

| | |
|---|---|
| Sample size | No statistical methods were used to predetermine sample sizes. Sample sizes for smFISH, IF, RT-qPCR, WB, developmental monitoring of embryos, EU incorporation assay, total RNA-seq and miniATAC-seq are consistent with current standards. |
| Data exclusions | No data were excluded from analyses. |
| Replication | We confirmed consistent results between three-to-six independent biological replicates (and/or experiments) for qPCR, smFISH and IF and EU incorporation assay. For Nextgen sequencing analysis, we also confirmed consistent results between two independent biological replicates for total RNA-seq and miniATAC-seq experiments based on Pearson's correlation coefficient of gene expressions and read enrichments, as determined by SeqMonk. The number of replicates/experiments has been described in the figures, legends and main text. |
| Randomization | The experiments were not randomized. Sample were allocated as either control or experimental (KD, CRISPRi and/or rescue) groups. |
| Blinding | The experiments were not blinding. However our analytical pipeline for each experiment followed uniform criteria applied to all samples, allowing us to analyze data, unbiasedly. |

# Reporting for specific materials, systems and methods

We require information from authors about some types of materials, experimental systems and methods used in many studies. Here, indicate whether each material, system or method listed is relevant to your study. If you are not sure if a list item applies to your research, read the appropriate section before selecting a response.

## Materials & experimental systems

| n/a | Involved in the study |
|---|---|
| ☐ | ☒ Antibodies |
| ☐ | ☒ Eukaryotic cell lines |
| ☒ | ☐ Palaeontology and archaeology |
| ☐ | ☒ Animals and other organisms |
| ☒ | ☐ Human research participants |
| ☒ | ☐ Clinical data |
| ☒ | ☐ Dual use research of concern |

## Methods

| n/a | Involved in the study |
|---|---|
| ☒ | ☐ ChIP-seq |
| ☒ | ☐ Flow cytometry |
| ☒ | ☐ MRI-based neuroimaging |

# Antibodies

| | |
|---|---|
| Antibodies used | The methods section of the manuscript contains information on all antibodies.<br>Mouse anti-MERVL Gag monoclonal antibody (A2 and C1, 1/5 dilution, generated in our lab (Guo Y., in prep))<br>Mouse anti-β-tubulin monoclonal antibody (E7, 1/2000 dilution, E7, Developmental Studies Hybridoma Bank)<br>Rabbit anti-OCT4 monoclonal antibody (EPR17929, 1/100 dilution, ab181557, abcam)<br>Mouse anti-CDX2 monoclonal antibody (CDX2-88, 1/100 dilution, ab157524, abcam)<br>Mouse anti-E-Cadherin monoclonal antibody (36, 1/100 dilution, 610182, BD Transduction Laboratories)<br>Rabbit anti-Cleaved Caspase-3 polyclonal antibodies (1/200, 9661, CST)<br>Rabbit anti-Phospho-S15-p53 polyclonal antibodies (1/200 dilution, 9284, CST)<br>HRP-conjugated goat anti-mouse IgG secondary antibody (1/10000 dilution, #330, MBL life science)<br>Alexa Fluor 488-conjugated goat anti-mouse IgG secondary antibody (1/500 dilution, A-11001, Thermo Fisher Scientific)<br>Alexa Fluor 568-conjugated goat anti-mouse IgG secondary antibody (1/500 dilution, A-11004, Thermo Fisher Scientific)<br>Alexa Fluor 488-conjugated goat anti-rabbit IgG secondary antibody (1/500 dilution, A-11008, Thermo Fisher Scientific)<br>Alexa Fluor 568-conjugated goat anti-rabbit IgG secondary antibody (1/500 dilution, A-11011, Thermo Fisher Scientific) |
| Validation | Antibodies used for immunofluorescence analysis and western blotting were validated by manufacturers (or us).<br>Mouse anti-MERVL Gag monoclonal antibody : in this study and Guo Y., in prep<br>Mouse anti-β-tubulin monoclonal antibody : https://dshb.biology.uiowa.edu/E7_2<br>Rabbit anti-OCT4 monoclonal antibody : https://www.abcam.co.jp/oct4-antibody-epr17929-chip-grade-ab181557.html<br>Mouse anti-CDX2 monoclonal antibody : https://www.abcam.co.jp/cdx2-antibody-cdx2-88-ab157524.html<br>Mouse anti-E-Cadherin monoclonal antibody : https://www.citeab.com/antibodies/2412208-610182-bd-transduction-laboratories-purified-mouse<br>Rabbit anti-Cleaved Caspase-3 polyclonal antibodies : https://www.cellsignal.jp/products/primary-antibodies/cleaved-caspase-3-asp175-antibody/9661<br>Rabbit anti-Phospho-S15-p53 polyclonal antibodies : https://www.cellsignal.jp/products/primary-antibodies/phospho-p53-ser15-antibody/9284 and Cui W. et al. Sci Rep., 2016 : https://www.nature.com/articles/srep37396<br>HRP-conjugated goat anti-mouse IgG secondary antibody : https://ruo.mbl.co.jp/bio/dtl/dtlfiles/330_v5.pdf<br>Alexa Fluor 488-conjugated goat anti-mouse IgG secondary antibody : https://www.thermofisher.com/antibody/product/Goat-anti-Mouse-IgG-H-L-Cross-Adsorbed-Secondary-Antibody-Polyclonal/A-11001<br>Alexa Fluor 568-conjugated goat anti-mouse IgG secondary antibody : https://www.thermofisher.com/antibody/product/Goat-anti-Mouse-IgG-H-L-Cross-Adsorbed-Secondary-Antibody-Polyclonal/A-11004<br>Alexa Fluor 488-conjugated goat anti-rabbit IgG secondary antibody : https://www.thermofisher.com/antibody/product/Goat-anti-Rabbit-IgG-H-L-Cross-Adsorbed-Secondary-Antibody-Polyclonal/A-11008<br>Alexa Fluor 568-conjugated goat anti-rabbit IgG secondary antibody : https://www.thermofisher.com/antibody/product/Goat-anti-Rabbit-IgG-H-L-Cross-Adsorbed-Secondary-Antibody-Polyclonal/A-11011 |

# Eukaryotic cell lines

Policy information about cell lines

| | |
|---|---|
| Cell line source(s) | Doxycycline (Dox)-inducible Dux ESC line, generated in our laboratory (Li T.D., et al. 2022) was used in this study. |
| Authentication | Since these cells were easily distinguished based on colony morphologies, cell lines have been authenticated by microscopic inspection. |
| Mycoplasma contamination | None of the cell lines used have been tested. |
| Commonly misidentified lines (See ICLAC register) | No commonly misidentified cell lines were used in this study. |

# Animals and other organisms

Policy information about studies involving animals; ARRIVE guidelines recommended for reporting animal research

| | |
|---|---|
| Laboratory animals | Wild-type BDF1 female and male mice at 56-84 days of age were used for the knockdown (KD) experiment. Mice were maintained and fed ad libitum with standard diet and water in a temperature-, humidity-, light-controlled room (23±3 degree Celsius, humidity of 50±10%, 14 light/10 dark cycle). |
| Wild animals | No wild animals were used in this study. |
| Field-collected samples | No field-collected samples were used in this study. |
| Ethics oversight | Mice were maintained and used according to the guidelines of the Institutional Animal Care and Use Committee (protocol no. 09105-(10) and 11045-(6)) at Keio University. |

Note that full information on the approval of the study protocol must also be provided in the manuscript.

