## [Peer Review File · Nature Genetics]

Peer Review Information

Manuscript Title: Transcription of MERVL Retrotransposons Is Required for Preimplantation Embryo Development

Corresponding author name(s): Haruhiko Siomi

Reviewer Comments & Decisions:

Decision Letter, initial version:

13th May 2022

Dear Haru,

Your Article, "Transcription of Murine Endogenous Retrovirus MERVL Is Required for Progression of Development in Early Preimplantation Embryos", has now been seen by 3 referees. I sincerely apologize for the long review process. One of the reviewers took longer than expected to submit their comments.

You will see from their comments copied below that while they think your findings are of considerable potential interest, they have raised substantial concerns regarding the appropriateness/robustness of the technical approach that must be thoroughly addressed. In light of these comments, we cannot accept the manuscript for publication, but would be interested in considering a revised version that addresses these serious concerns.

Reviewer #1 is worried that the findings may be an artifact of ASO technology. They have therefore asked for some additional controls. If the results are robust, the reviewer thinks they are important. Reviewer #2 finds the paper interesting but preliminary. Their main concerns are about staging (comparison between KD and WT embryos) and the level of mechanistic insight (similar point to Reviewer #1's).

Reviewer #3 also thinks that the results are potentially important/exciting. However, like Reviewer #1, they would want to see more controls/approaches to make sure that the ASO-based data are robust.

We hope you will find the referees' comments useful as you decide how to proceed. If you wish to submit a substantially revised manuscript, please bear in mind that we will be reluctant to approach the referees again in the absence of major revisions.

If you choose to revise your manuscript taking into account all reviewer comments, please highlight all changes in the manuscript text file. At this stage we will need you to upload a copy of the manuscript in MS Word .docx or similar editable format.

We are committed to providing a fair and constructive peer-review process. Do not hesitate to contact me if there are specific requests from the reviewers that you believe are technically impossible or unlikely to yield a meaningful outcome. I would be happy to discuss the reviewers' comments in detail.

*2) If you have not done so already please begin to revise your manuscript so that it conforms to our Article format instructions, available [here](http://www.nature.com/ng/authors/article_types/index.html). Refer also to any guidelines provided in this letter.

[redacted]

If you wish to submit a suitably revised manuscript we would hope to receive it within 6 months. If you cannot send it within this time, please let us know. We will be happy to consider your revision so long as nothing similar has been accepted for publication at Nature Genetics or published elsewhere. Should your manuscript be substantially delayed without notifying us in advance and your article is eventually published, the received date would be that of the revised, not the original, version.

Thank you for the opportunity to review your work.

Sincerely,

Tiago

Tiago Faial, PhD
Senior Editor
Nature Genetics
<https://orcid.org/0000-0003-0864-1200>

Reviewers' Comments:

Reviewer #1:

Remarks to the Author:

The Sakashita/Kitano et al. manuscript explores a fundamental unanswered question in early development, namely—do highly expressed, stage-specific repetitive elements provide beneficial functions for the host, and if so, how?

The major claims of the manuscript:

- 1) By using smFISH, the authors find that MERVL is activated earlier than previously reported (via poly-A primed SMRT-seq) by others and it curiously retained in the nucleus in zygotes and early 2-cell stage embryos. This RNA expression precedes the accumulation of MERVL GAG protein at the mid-2-cell stage. (Figure 1)
- 2) By using the antisense oligo (ASO) technology, they were able to deplete MERVL RNA from the embryo, and this treatment largely prevents the timely development of the embryo and the formation of blastocysts (Fig 2).
- 3) However, the depletion of MERVL RNA in the cytoplasm and MERVL GAG protein using siRNA technology does not impair development. This strong phenotype seen in Fig 2 with the ASO depletion of MERVL cannot be rescued by using ASO-resistant MERVL mRNA injection in trans, which indicates that the MERVL gene products (mRNA/protein) in the cytoplasm are not sufficient to ablate the effect of the ASO. (Fig 3).
- 4) In Fig 4 they show MERVL ASO-depleted embryos have molecular defects largely characterized by failure to exit the transcriptional program of the 2-cell stage, and concomitantly exhibit accumulation of phospho-p53 indicating that signaling (typically through the DDR on S15) is activated. (Fig 4).
- 5) In figure 5 they go on to show that the MERVL-ASO depleted embryos also exhibit a developmental delay in decommissioning some of the 2-cell stage specific open chromatin regions, consistent with the transcriptional effects seen in the previous figure (Fig 5).

In general, the paper was a joy to read, the figures were presented clearly, it was well-referenced, and I think it will be of great interest to those in both the TE community and the early embryo developmental biologists. Resolving the true function of MERVL in development and ZGA is an important goal for the field.

Major criticism:

It is nice that the authors can parse out effects of the MERVL-transcription or MERVL-nascent transcriptions from the MERVL gene products (mRNA/protein) in the cytoplasm. However, I have strong concerns about how the ASO experiments were performed. Additional control experiments would largely alleviate these concerns. For instance, even after reading their methods, it is unclear what the control for these ASO experiments might be. Is it a mock injected or scrambled ASO? Why not use a MERVL sense-oligo as a control, since this should not affect the accumulation of nascent MERVL transcripts?

How do we know that flooding the embryo with MERVL ASO does not profoundly interfere with DNA replication? Presumably these ASO could also base-pair with MERVL sense strand during DNA replication and since the ASO do not likely contain 5'-phosphates, this could result in nicks of the DNA that would need to be sealed by the cellular DNA kinases/ligases. A typical control for this possibility is a sense oligo that eliminates the possibility of the experimental reagent perturbing DNA replication. Some applications of ASO technology to non-repetitive sequences (single or low-copy genes) may not be prone to this type of technical complication, but since there are ~600 MERVL instances across the genome, it is important to carefully consider this possibility. Since the authors detect strong phospho-p53 accumulation in ASO-injected embryos (Fig 4J) I wonder if this is caused by ASO interference with DNA replication.

My other major criticism is a lack of substantive mechanism that described how MERVL transcripts (presumably in the nucleus) or the act of MERVL transcription itself leads to the strong embryo phenotype. If the phenomenon is not an artifact of ASO technology (I would love to be wrong about this), how do the authors propose that this leads to the developmental delay seen? Are transcribed MERVL loci in the nucleus spatially clustered into transcription factories that allow for more robust ZGA genome-wide? Does the chromatin activation of MERVL copies at ZGA allow for a reprogramming similar to what was reported for LINE-1 in Jachowicz et al. Nature Genetics 2017?

Minor comments:

1. Would like to see Fig 1D quantified, potentially even with higher resolution single-molecule counting. Is there any difference between paternal and maternal pronuclei and their MERVL expression in Fig 1D?
2. Fig 2F images are low-quality/low-resolution. If the purpose of the figure panel is to count the nuclei that are OCT4+ or CDX2+ this will suffice but these images are not publication-quality.
3. Fig 3F: why if the MERVL mRNA is injected into the pronuclei of the zygote is it exclusively retained in the nucleus of the 2-cell embryo even after nuclear-envelope breakdown and mitosis? Is there an active transport mechanism into the nucleus or does the MERVL mRNA stay attached to mitotic chromosomes and thus retained in the nucleus?
4. In Fig 4J: have the authors considered using some of the well-characterized small-molecule inhibitors of the kinases in the DDR signaling pathway (or instead using siRNA to Trp53) to test if the prolonged 2-cell program involves p53, or is phospho-p53 not a driver but a consequence of MERVL-depletion? In our hands, even ATR/ATM/CHECK1/CHECK2 inhibition in embryo culture does not significantly perturb development to the blastocyst stage—although by ablating checkpoints the embryos are likely to have aneuploidies. Does the ASO itself trigger a single-stranded DNA damage response through ATR and activate p53?
5. Fig 4E: would be more standard to separate this analysis into all genes and non-maternal (exclusively zygotically expressed) transcripts. Since ZGA is intimately intertwined with gene products that help to clear the maternally deposited transcripts, it can be difficult to judge from this type of combined analysis

if the gene expression changes depicted are truly ZGA-exclusive genes or dominated by maternal mRNA clearance.

6. Apart from the genome browser screen-shots in Fig 4i (which has no label on the y-axis for FPKM, etc.), I would like to see some more quantitative depictions of a panel of genes that fail to shut off in the ASO-MERVL embryos after the 2-cell stage.

7. Aggregate plots in Fig 4h are not preferred for these types of data as they obscure the true granularity and/or heterogeneity amongst individual loci within a grouping (cluster). I like that these plots have statistical tests applied, but it would be nice to see how much heterogeneity there is, potentially with a heatmap. Are these patterns driven by a large subset of regions within the cluster or are there outliers that dominate the aggregate pattern?

8. On line 86-88, the authors state that “In addition, the maternal and zygotic knockout of Dux, an upstream regulator of MERVL, was tolerated and ZGA and preimplantation development proceeded as normal”. While loss of DUX is not completely incompatible with ZGA or development, the overall statement is not accurate. The Iaco et al. Development paper from 2020 paper shows that ~50% of the DUX-KO embryos arrest at early cleavage stages while the Chen, et al. Nature Genetics 2019 paper reports a 50% reduction in litter size in DUX-KO mice. I would hardly agree that 50% death in the offspring is “preimplantation development proceeding as normal” as defects in these early stages can often manifest after implantation.

9. Lines 472-474 states “Grow et al. recently demonstrated that chemically induced DNA damage invoked a DNA damage response (DDR) which in turn led to the activation of p53 followed by 2C genes” but this paper also used a rare-cutting endonuclease to create dsDNA breaks and activate DUX/MERVL indicating that DNA damage per se apart from chemical treatment is capable of recapitulating this effect.

10. Fig 1a cartoon depicts pluripotency arising at the 4-cell stage. Most mouse embryologists would argue that this is the “pre-lineage” stage and pluripotency only is formed up to the early blastocyst stage, see Posfai, et al. eLIFE 2017.

11. Fig 3a: “nucleas” (sic) instead of nucleus, “degredate” (sic) instead of degrade.

Reviewer #2:

Remarks to the Author:

In this paper, Sakashita et al. investigate the role that MERVL – a transposable element with over a thousand of copies in the mouse genome – plays during zygotic genome activation and pre-implantation development. By using a novel ASO-based strategy for knockdown, the authors demonstrate that loss of MERVL leads to delayed, and eventually halted, development. Through the combination of siRNA-based knockdown and trans-expression rescue experiments, Sakashita et al. conclude that this critical role of MERVL stems from either the cis-acting nascent RNA or transcription of the MERVL locus. RNA-seq and ATAC-seq analyses were subsequently used to conclude that loss of MERVL leads to prolonged

expression of 2-cell stage genes, which is associated with a prolonged state of chromosomal openness surrounding 2-cell genes. The authors claim that their data suggests a critical role for MERVL in regulating the totipotency to pluripotency transition, as well as the ensuing differentiation and development of the pre-implantation embryo.

We find the strategies used here to investigate MERVL function to be novel and intriguing, although we do have some concerns about the strength of the claims that the authors make regarding MERVL and the ways in which it modulates pre-implantation development that should be addressed prior to publication.

Major concerns:

1. For all comparisons of gene expression between control and MERVL-KD embryos, it is unclear how the authors staged their samples. Were embryos visually assessed to confirm that those collected for the control and KD samples were of similar stages? If not, all MERVL-KD samples would be composed of a mixture of embryos at a variety of different stages, with the majority stalled at an earlier stage in development. This would lead to differences in lineage markers (Fig. 2h), differences in 2-cell stage gene expression (Fig. 4g-h), and differences in chromatin openness at these 2-cell stage genes (Fig. 5c). To ensure that the claims the authors make about differences in gene expression are not influenced by inappropriate staging, this issue must be addressed explicitly in both the text and the methods section.
2. Although the authors claim that the requirement for MERVL is due to its cis-acting function or because of its transcription, it's unclear how either of these functions would lead to the upregulation of 2-cell stage genes observed in this paper. The authors should more thoroughly discuss the mechanism of MERVL action. Are dysregulated genes more likely to be in close proximity to a MERVL element? Are the cohort of genes next to MERVL elements more likely to be dysregulated? It seems as though for the mechanism that the authors propose to be true, there needs to be some sort of relationship between proximity to MERVL and dysregulation.
3. A key point of this paper, is that the authors claim that their method of knocking down MERVL is superior to previous methods, thus revealing the true effect of loss of MERVL on development. To support this claim, the authors should more thoroughly address the differences between their technique and other studies earlier on in their paper. It is concerning that, even with the control ASO injection, as many as 30.9% of embryos stall at the 2-4 cell stage. This suggests that either the injection procedure or introduction of any ASO is associated with some degree of lethality, and raises concerns about the extent to which ASO injection (and not MERVL KD) could be contributing to the strong phenotypes observed for the MERVL ASO KD. For injection of the siRNA, was embryo stalling and lethality equally as high as for the control ASO? Additionally, for studies that have been done with knockout mice in which transcriptional regulators of ZGA have been perturbed (such as *Dppa2/4*), were levels of MERVL assessed?

Minor concerns:

1. Could the authors discuss more why MERVL begins to be transcribed at the zygote stage, but doesn't appear in the cytoplasm until the mid 2-cell stage (Fig. 1d)? Is there some sort of delay in RNA export from the nucleus at that developmental time?
2. There seems to be heterogeneity in MERVL levels (as determined by smFISH) between nuclei in the 4-cell embryo (Fig. 1d). Is this a general trend across all images examined, and if so, what might the consequences of such heterogeneity be?
3. The key for Fig. 2d lists both 8-64 cell embryos as well as morula stage embryos. The morula stage begins at 16 cells, and thus these two classifications overlap with each other. Please update the legend (as well as the analyses) to follow this nomenclature.
4. A major conclusion of this work is that the requirement for MERVL is due to either cis-acting properties, or the act of transcription of the transposable element. To make this conclusion, Sakashita et al. rely on comparisons between ASO-mediated knockdown, miRNA-mediated knockdown, and ASO-mediated knockdown with rescue of MERVL in trans. To strengthen these conclusions, it would be nice if the authors showed, for all treatments, both the MERVL RNA levels and localization (using smFISH) in addition to the MERVL Gag protein levels at the early and late 2-cell stage. This would allow readers to understand the extent to which cytoplasmic MERVL RNA levels are altered in each scenario.
5. For all figures showing genome browser tracks (Fig. 4i, Fig. 5d) for specific genes, some sort of scale bar or axis should be used to specify the relative heights of each of the tracks.

Reviewer #3:

Remarks to the Author:

The authors report that the RNA produced at MERVL endogenous retroviruses play essential roles during early stages of mammalian development. The authors developed an ASO-mediated "knockdown" system for MERVL RNA, and found that ASO-mediated MERVL knockdown in blastocysts arrested development at ~2.5 days. siRNA-mediated knockdown of MERVL RNA did not arrest development, suggesting that nuclear presence MERVL RNA is essential for its function. Injection of MERVL RNA did not rescue the developmental arrest in MERVL-knockdown blastocysts (using the ASO-approach), suggesting that the RNAs act in cis. The authors characterized transcriptome and chromatin accessibility changes in MERVL-knockdown blastocysts. These data revealed that transcription of fusion transcript driven by MERVL LTR elements did not change, further supporting that it is the MERVL RNA in cis that is essential for developmental progression. The transcriptome and chromatin accessibility of the MERVL-knockdown cells were similar to those of 2C-like cells, consistent with developmental arrest.

The manuscript provides interesting and important new insights into the roles of endogenous retroviruses during early development. Endogenous retroviruses (ERVs) are prevalent ancient transposable elements in the mammalian genomes. Much of the importance of ERVs to cellular functions has been attributed to their Long Terminal Repeats (LTRs) encoding transcription factor

binding sites, and acting as promoters and enhancers of cellular genes (e.g. Macfarlan et al., Nature 2012). This manuscript provides evidence that the RNA species produced at MERVL ERVs have critical regulatory roles in early murine embryos, shedding new light into how retrotransposons may affect critical cellular functions.

The data in the manuscript appear of high quality and appropriate for supporting the authors conclusions. I only have one major and a few minor suggestions for improvements.

Major comment

The main conclusions of the manuscript rely on ASO-mediated knockdown of MERVL RNA. The authors performed substantial bioinformatic analyses and control experiments to validate the ASOs they used. However, ASO-mediated knockdown is generally prone to artefacts since the pathways that lead to RNA degradation are incompletely understood. Since the key insights of the manuscript hinge on specificity of the ASO-knockdown, further controls or an alternative approach is highly recommended. For example, the authors could design and test various other sets of ASOs that target non-overlapping regions and test the existing and new ASOs against a larger set of control ASO including more scrambled controls. Also, the CasRX system (Konnerman et al., Cell 2018) could be used as an ASO-independent knockdown approach that works in cis to validate the ASO-results. shRNAs are also known to be processed and mediate knockdown in the nucleus to some degree.

Minor comments

The authors call MERVL RNA species (and RNA FISH assay) “nascent”, but then explain that MERVL RNAs are intronless. Since the term ‘nascent’ typically refers to unprocessed, intron-containing RNAs, clarification of the terminology is recommended.

Some of the transcriptomic changes highlighted by the authors seem minor. To help readers gauge extent, please add y-axes with units to the displayed Seq data (e.g. Fig 4i, Fig 5d, Supplementary Fig 5c).

Author Rebuttal to Initial comments

GENERAL COMMENTS FOR ALL REVIEWERS

We thank the reviewers for their kind and thoughtful consideration of our manuscript, which have allowed us to considerably improve the manuscript within a reasonable amount of time. All reviewers are positive about the potential interest of our current study and have requested other controls/approaches to prove robustness and reproducibility of the embryonic phenotypes upon ASO-mediated KD of MERVL. We sincerely appreciate the specific points raised by the

reviewers and find that it is feasible to thoroughly address these points in the revision of this work. Accordingly, in this revised manuscript, we newly perform (1) microinjection of sense-oligos (SOs) which are complementary to the individual MERVL-targeting ASO into the zygotes, as a control for ASO-mediated KD, (2) CasRx-mediated KD of MERVL and (3) CRISPRi-mediated repression of MERVL, as other approaches to remove or repress MERVL transcripts in preimplantation development (whole new figures are generated for these points as New Figure 3, Supplementary Figure S2, S3 and S6). We **conclude that developmental phenotypes, observed in ASO-mediated MERVL-KD embryos are robust.** In addition, to address major comment 3 of reviewer #1 and comment 2 of reviewer #2, we have also reanalyzed our RNA-seq data to evaluate how MERVL transcripts regulate gene expression and chromatin remodeling in *cis* (New Figure 6). These new results further support our original conclusion.

Point-by-Point RESPONSE TO REVIEWERS

Reviewer #1:

In general, the paper was a joy to read, the figures were presented clearly, it was well-referenced, and I think it will be of great interested to those in both the TE community and the early embryo developmental biologists. Resolving the true function of MERVL in development and ZGA is an important goal for the field.

We thank the reviewer for acknowledging the important contribution of our work to the field of transposon and early embryo development research. This is very important, we agree.

Major criticism:

1. It is nice that the authors can parse out effects of the MERVL-transcription or MERVL-nascent transcriptions from the MERVL gene products (mRNA/protein) in the cytoplasm. However, I have strong concerns about how the ASO experiments were performed. Additional control experiments would largely alleviate these concerns. For instance, even after reading their methods, it is unclear what the control for these ASO experiments might be. Is it a mock injected or scrambled ASO? Why not use a MERVL sense-oligo as a control, since this should not affect the accumulation of nascent MERVL transcripts?

We thank reviewer #1 for his/her careful consideration of our manuscript and for constructive criticisms and suggestions. Because we had only performed microinjection of randomized non-targeting ASO for the control in the original version of our manuscript (we now added this information in the Method section clearly (Page 21 Line 735~), the comments therefore led us to newly perform microinjection of sense-oligos (SOs) that are complementary to respective ASOs against MERVL, as the control for ASO experiments. Based on this new experiment, we confirmed that microinjection of SOs (individual and mixed) to zygotes did not affect the expression level of MERVL transcript and preimplantation development, corroborating that the phenotype of ASO-mediated MERVL-KD embryos is unlikely to arise from an artificial effect by ASO injection. We now include the data in the text (Page 5 Line 174~) and figure (New Supplementary Figure S2).

2. How do we know that flooding the embryo with MERVL ASO does not profoundly interfere with DNA replication? Presumably these ASO could also base-pair with MERVL sense strand during DNA replication and since the ASO do not likely contain 5'-phosphates, this could result in nicks of the DNA that would need to be sealed by the cellular DNA kinases/ligases. A typical control for this possibility is a sense oligo that eliminates the possibility of the experimental reagent perturbing DNA replication. Some applications of ASO technology to non-repetitive sequences (single or low-copy genes) may not be prone to this type of technical complication, but since there are ~600 MERVL instances across the genome, it is important to carefully consider this possibility. Since the authors detect strong phospho-p53 accumulation in ASO-injected embryos (Fig 4J) I wonder if this is caused by ASO interference with DNA replication.

We appreciate this critical question. We agree that it is important to exclude the possibility that the phenotypes of ASO-mediated MERVL-KD embryos arise from an artificial effect by ASO injection. As mentioned above, in this revised manuscript, we newly perform microinjection of SOs that are complementary to respective ASOs against MERVL, as the control for ASO experiments. We show no discernible defects in SO-injected embryos both at the expression level of MERVL transcript and developmental competence (New Supplementary Figure S2), indicating that the phenotype of ASO-mediated MERVL-KD embryos is unlikely to arise from artifacts, such as DNA replication stress and innate immune response. In addition, we also found significantly higher enrichment of phospho-p53 in ASO-mediated MERVL-KD embryos at all developmental stages we examined, compared to that in SO-induced embryos, further supporting the notion that accumulation of phospho-p53 in MERVL-KD embryos is unlikely to arise from artifacts by injection of small oligonucleotides against MERVL (Additional Review Material 1). We now include the data in the text (Page 5 Line 174~) and figure (New Supplementary Figure S2).

Additional Review Material 1. Immunofluorescence analysis of nuclear p53 levels in MERVL SO- and ASO-injected embryos.
 (a) Representative images of immunofluorescence staining with antibody raised against phosphorylated p53 on serine 15 (p-S15-p53) with the DNA counterstained with DAPI in MERVL SO- and ASO-induced embryos at 2-cell, 4-cell and 8-cell stages. Scale bar: 20 μ m.
 (b) Dot plots showing the relative intensity for p-S15-p53 per nucleus in MERVL SO- and ASO-injected embryos at 2-cell, 4-cell and 8-cell stages, normalized to blank value. * P <0.05, *** P <0.001, two-tailed unpaired t -tests.

3. My other major criticism is a lack of substantive mechanism that described how *MERVL* transcripts (presumably in the nucleus) or the act of *MERVL* transcription itself leads to the strong embryo phenotype. If the phenomenon is not an artifact of ASO technology (I would love to be wrong about this), how do the authors propose that this leads to the developmental delay seen? Are transcribed *MERVL* loci in the nucleus spatially clustered into transcription factories that allow for more robust ZGA genome-wide? Does the chromatin activation of *MERVL* copies at ZGA allow for a reprogramming similar to what was reported for *LINE-1* in Jachowicz et al. *Nature Genetics* 2017?

We appreciate this important point and now investigate the regulatory link between *MERVL* transcription and host preimplantation development, through re-analysis of our RNA-seq data in ASO-mediated *MERVL*-KD embryos (New Figure 6). To this end, we firstly focus on whole transcriptome including transcripts from annotated RefSeq genes and unannotated transcriptionally active regions (uTAR) in scrambled control and *MERVL*-KD embryos. Subsequently, we detected 3,499 differentially expressed uTAR between scrambled control and *MERVL*-KD embryos; in particular, the vast majority of them (3,065/3,499, 87.6%) are shown to be downregulated upon the loss of *MERVL* transcription (New Figure 6b, d-f). Specifically, we find a higher fraction of differentially expressed uTARs preferentially located in the neighborhood of *MERVL* (New Figure 6g), while no positive correlation is detected between expression changes for annotated RefSeq genes upon *MERVL*-KD and their proximity to *MERVL* (New Figure 6a). Moreover, concomitantly with the change in expression of uTARs upon *MERVL*-KD, the intensity of ATAC-seq signal on differentially expressed uTARs is also decreased in *MERVL*-KD embryos (New Figure 6h). These analyses have led to an important conclusion that KD of *MERVL* transcript repress not only transcription of *MERVL* itself but also adjacent transcription and chromatin

accessibility up to 50 kb upstream and downstream of MERVL, suggesting that *en masse* activation of interspersed MERVL copies at ZGA can drive genome-wide reprogramming towards the onset of ontogeny, in contrast to what was reported for LINE1 in which only modest change in gene expression was detectable in 2-cell stage embryos upon targeted repression of LINE1 (Jachowicz JW. *et al.* 2017). We now include this data in the text (Page 14 Line 492~, P19, Line 678~) and in New Figure 6 with appropriate citations.

Minor comments:

1. Would like to see Fig 1D quantified, potentially even with higher resolution single-molecule counting. Is there any difference between paternal and maternal pronuclei and their MERVL expression in Fig 1D?

As suggested by the reviewer, we quantified the intensity of MERVL transcript in the male and female pronuclei of zygote and confirmed that no difference was observed in the expression of MERVL transcript between the male and female pronuclei of zygote (Additional Review Material 2). We have added this notion to the text (Page 4, Line 129~).

Additional Review Material 2. smFISH for MERVL transcript in PN stage zygotes.

(a) Representative image of smFISH with probes set for full-length MERVL RNA sequence with the DNA counterstained with DAPI in PN stage zygotes. Scale bar: 20 μ m. ♀, female PN; ♂, male PN. (b) Dot plot showing the relative intensity for MERVL transcript, normalized to DAPI signal, in male and female PN of the zygotes. N.S., not significant, two-tailed unpaired *t*-tests.

2. Fig 2F images are low-quality/low-resolution. If the purpose of the figure panel is to count the nuclei that are OCT4+ or CDX2+ this will suffice but these images are not publication-quality.

We thank the reviewer for pointing this out. We replaced images with new ones with high resolutions.

3. Fig 3F: why if the MERVL mRNA is injected into the pronuclei of the zygote is it exclusively retained in the nucleus of the 2-cell embryo even after nuclear-envelope breakdown and mitosis? Is there an active transport mechanism into the nucleus or does the MERVL mRNA stay attached to mitotic chromosomes and thus retained in the nucleus?

We agree that it is interesting to elucidate how MERVL transcript is retained in the nucleus of early 2-cell embryos after first cleavage. At this time, we speculate that MERVL transcript may interact with other RNA binding protein partners to reside in the nucleus. For example, there is evidence that transcript from LINE1 act as chromatin-associated RNA that binds Nucleolin, Kap1 and chromatin in regulating gene expression for the self-renewal of ES cells (Percharde, M. *et al.* 2018). Clarification of the potential mechanism for nuclear localization of MERVL transcript warrants our future investigation. We have added this notion to the text (Page 18, Line 641~) with appropriate citations.

4. In Fig 4J: have the authors considered using some of the well-characterized small-molecule inhibitors of the kinases in the DDR signaling pathway (or instead using siRNA to Trp53) to test if the prolonged 2-cell program involves p53, or is phospho-p53 not a driver but a consequence of MERVL-depletion? In our hands, even ATR/ATM/CHECK1/CHECK2 inhibition in embryo culture does not significantly perturb development to the blastocyst stage—although by ablating checkpoints the embryos are likely to have aneuploidies. Does the ASO itself trigger a single-stranded DNA damage response through ATR and activate p53?

Related to major criticism #2; as mentioned above, we now found significantly higher enrichment of phospho-p53 in ASO-injected MERVL-KD embryos at all developmental stages we examined compared to that in SO-injected embryos, confirming that accumulation of phospho-p53 in MERVL-KD embryos is unlikely to arise from artifacts by injection of small oligonucleotides against MERVL (Additional Review Material 1).

5. Fig 4E: would be more standard to separate this analysis into all genes and non-maternal (exclusively zygotically expressed) transcripts. Since ZGA is intimately intertwined with gene products that help to clear the maternally deposited transcripts, it can be difficult to judge from this type of combined analysis if the gene expression changes depicted are truly ZGA-exclusive genes or dominated by maternal mRNA clearance.

As suggested, we further classified DEGs upon MERVK-KD into “maternally inherited genes” and “zygotic genes” based on the DBTMEE v2 database. We found that only 3-15% of genes were defined as “maternally inherited RNA” across all stages examined (New Figure 4f), suggesting that MERVL-KD led to major defects in expression of zygotic genes, rather than maternal mRNA clearance. We now included this data in the text (Page 10 Line 333~) and in New Figure 4f.

6. *Apart from the genome browser screen-shots in Fig 4i (which has no label on the y-axis for FPKM, etc.), I would like to see some more quantitative depictions of a panel of genes that fail to shut off in the ASO-MERVL embryos after the 2-cell stage.*

According to the reviewer’s comment, we newly performed unbiased hierarchical clustering analysis of all 2C gene transcripts (n = 1,769) from RNA-seq data in control and MERVL-KD embryos, revealing that more than half of 2C genes were dysregulated in at least one developmental stage of MERVL-KD embryos (New Supplementary Figure S4d). We now included this data in the text (Page 11 Line 373~) and in New Supplementary Figure S4d. It further supports our conclusion. Besides, we also added labels on the y-axis for the normalized read count, more clearly to representative track views (in Figure 4i and 5d, New Figure 6f, New Supplementary Figure S6f and S7c)

7. *Aggregate plots in Fig 4h are not preferred for these types of data as they obscure the true granularity and/or heterogeneity amongst individual loci within a grouping (cluster). I like that these plots have statistical tests applied, but it would be nice to see how much heterogeneity there is, potentially with a heatmap. Are these patterns driven by a large subset of regions within the cluster or are there outliers that dominate the aggregate pattern?*

Related to minor comment #6: we thank the reviewer for pointing this out. We have removed old inappropriate Figure 4h and added new data for unbiased hierarchical clustering analysis of all 2C gene transcripts (n = 1,769) to New Supplementary Figure S4d.

8. *On line 86-88, the authors state that “In addition, the maternal and zygotic knockout of Dux, an upstream regulator of MERVL, was tolerated and ZGA and preimplantation development proceeded as normal”. While loss of DUX is not completely incompatible with ZGA or development, the overall statement is not accurate. The Iaco et al. Development paper from 2020 paper shows that ~50% of the DUX-KO embryos arrest at early cleavage stages while the Chen, et al. Nature Genetics 2019 paper reports a 50% reduction*

in litter size in DUX-KO mice. I would hardly agree that 50% death in the offspring is “preimplantation development proceeding as normal” as defects in these early stages can often manifest after implantation.

We thank the reviewer for pointing this out. To state our point more accurately, we have revised this sentence to read; “In addition, the maternal and zygotic knockout of Dux, an upstream regulator of MERVL, partially impaired ZGA and preimplantation development, indicating that DUX plays a critical role, but not be essential for embryogenesis” (Page 3 Line 88~) with appropriate citations.

9. Lines 472-474 states “Grow et al. recently demonstrated that chemically induced DNA damage invoked a DNA damage response (DDR) which in turn led to the activation of p53 followed by 2C genes” but this paper also used a rare-cutting endonuclease to create dsDNA breaks and activate DUX/MERVL indicating that DNA damage per se apart from chemical treatment is capable of recapitulating this effect.

We thank the reviewer for the comment. To state our point more accurately, we have revised this sentence to read; “Grow et al. recently demonstrated that chemical and enzymatic induction of DNA damage invoked a DNA damage response (DDR) which in turn led to the activation of p53 followed by 2C genes” (Page 19 Line 651~).

10. Fig 1a cartoon depicts pluripotency arising at the 4-cell stage. Most mouse embryologists would argue that this is the “pre-lineage” stage and pluripotency only is formed up to the early blastocyst stage, see Posfai, et al. eLIFE 2017.

We thank the reviewer for the comment. According to the paper the reviewer cited and current knowledge in this field, we have changed this cartoon to an appropriate one.

11. Fig 3a: “nucleas” (sic) instead of nucleus, “degredate” (sic) instead of degrade.

Thank you ! We have corrected.

Reviewer #2:

We find the strategies used here to investigate MERVL function to be novel and intriguing, although we do have some concerns about the strength of the claims that the authors make regarding MERVL and the ways in which it modulates pre-implantation development that should be addressed prior to publication.

We thank the reviewer for the positive and encouraging comments. In accordance with the reviewer's suggestion, we have prepared the revised manuscript.

Major concerns:

1. For all comparisons of gene expression between control and MERVL-KD embryos, it is unclear how the authors staged their samples. Were embryos visually assessed to confirm that those collected for the control and KD samples were of similar stages? If not, all MERVL-KD samples would be composed of a mixture of embryos at a variety of different stages, with the majority stalled at an earlier stage in development. This would lead to differences in lineage markers (Fig. 2h), differences in 2-cell stage gene expression (Fig. 4g-h), and differences in chromatin openness at these 2-cell stage genes (Fig. 5c). To ensure that the claims the authors make about differences in gene expression are not influenced by inappropriate staging, this issue must be addressed explicitly in both the text and the methods section.

We thank the reviewer for his/her thoughtful and constructive criticisms and suggestions to improve our manuscript. For first comment raised by the reviewer, our response is "Yes", since we visually assessed and collected embryos at 2-cell, 4-cell and 8-cell stages under the stereomicroscope, thus confirming that the control and MERVL-KD samples were from the same cleavage stages. Thank you for pointing this out! We now added this notion to the text (Page 9, Line 321~, Page 25 Line 897~ and Page 26 Line 909~).

2. Although the authors claim that the requirement for MERVL is due to its cis-acting function or because of its transcription, it's unclear how either of these functions would lead to the upregulation of 2-cell stage genes observed in this paper. The authors should more thoroughly discuss the mechanism of MERVL action. Are dysregulated genes more likely to be in close proximity to a MERVL element? Are the cohort of genes next to MERVL elements more likely to be dysregulated? It seems as though for the mechanism that the authors propose to be true, there needs to be some sort of relationship between proximity to MERVL and dysregulation.

We appreciate this important point. We have provided new insight into the relationship between MERVL transcription and host gene expression, by re-analysis of our RNA-seq data in ASO-mediated MERVL-KD embryos (New Figure 6). To evaluate whether transcriptional dysregulation of genes upon MERVL-KD could be correlated with their proximity to MERVL, we firstly focused on whole transcriptome including transcripts from annotated RefSeq genes and unannotated transcriptionally active regions (uTAR) in control and MERVL-KD embryos. Interestingly, although no significant differences were observed between control and MERVL-KD embryos in the expression of MERVL-adjacent annotated genes, ranging from 0 to 500 kb in GENCODE vM25 (New Figure 6a), we detected 3,499 differentially expressed uTAR that preferentially located in the neighborhood of MERVL, between control and MERVL-KD embryos (New Figure 6d-g). Moreover, concomitantly with the change in expression of uTARs upon MERVL-KD, the intensity of ATAC-seq signal on differentially expressed uTARs was also decreased in MERVL-KD embryos (New Figure 6h). These analyses led to an important conclusion that KD of MERVL transcript prevented not only transcription of MERVL itself but also adjacent transcription and chromatin accessibility, suggesting that *en masse* activation of interspersed MERVL copies at ZGA can drive genome-wide reprogramming. We now included this data in the text (Page 14 Line 492~, Page 19, Line 678~) and in New Figure 6 with appropriate citations.

3. A key point of this paper, is that the authors claim that their method of knocking down MERVL is superior to previous methods, thus revealing the true effect of loss of MERVL on development. To support this claim, the authors should more thoroughly address the differences between their technique and other studies earlier on in their paper.

As suggested by the reviewer, we now include more information regarding the differences in KD strategy between present and previous studies in the text (Page 16, Line 575~ and Page 17, Line 579~) with appropriate citations.

Additionally, siRNA-KD and *in trans* RNA rescue assay suggest that *cis*-regulatory activity of MERVL loci plays a key role in host chromatin remodeling at totipotent-to-pluripotent transition. As a proof-of-concept for this hypothesis, in this revised manuscript, we now also performed CRISPRi-mediated repression of MERVL with dCas9-KRAB-MeCP2 and multiple-targeting gRNA constructs. This revealed that MERVLi embryos partially mimicked ASO-mediated KD of MERVL with regard to developmental defects and 2C transcriptome, although expression of MERVL was not fully repressed, probably due to inefficient expression of dCas9-KRAB-MeCP2 at zygote-to-early 2-cell stage in our system (New Figure 3i-l). These data support a MERVL *in cis* function that contributes to host chromatin remodeling. We have also included this data in the text (Page 8, Line 281~, Page 11 Line 392~ and Page 17, Line 596~) and in New Figure 3i-l and New Supplementary Figure S6 with appropriate citations.

It is concerning that, even with the control ASO injection, as many as 30.9% of embryos stall at the 2-4 cell stage. This suggests that either the injection procedure or introduction of any ASO is associated with some degree of lethality, and raises concerns about the extent to which ASO injection (and not MERVL KD) could be contributing to the strong phenotypes observed for the MERVL ASO KD.

We thank the reviewer for the comment. As the reviewer states above, at 2.5 dpc. 30.9% of control embryos remained at 2- to 4-cell stages. However, we do not think that development of these control embryos was stalled/arrested, because there is evidence that ~90% of the control embryos reached the blastocyst stage at 4.5 dpc, as being similar to normal developmental rate in vitro culture in KSOM (Gelber K. *et al.* 2011). We now added this notion to the text (Page 6, Line 192~). Importantly, in this revision, we are also able to disprove the possibility that the reduction of MERVL transcript level and aberrant developmental phenotypes in MERVL-KD embryos might arise from unexpected artificial effects due to the presence of excessive small nucleotides, such as DNA replication stress and innate immune response. To exclude this possibility, we performed microinjection of SOs which are complementary to the individual MERVL-targeting ASOs into zygotes, confirming that injection of SOs did not affect the expression of MERVL and preimplantation development (New Supplementary Figure S2), corroborating the notion that the phenotype of ASO-mediated KD embryos is unlikely to arise from an artificial effect. We now included this data in the text (Page 5 Line 174~) and in New Supplementary Figure S2.

For injection of the siRNA, was embryo stalling and lethality equally as high as for the control ASO?

Upon siRNA-mediated KD, the developmental rate to blastocyst stage is lower than those of ASO-control embryos (only 65.7% of control embryos reached the blastocyst stage), suggesting that siRNA-mediated KD is more genetically toxic than ASO-mediated one. However, given the observation that siRNA-mediated MERVL-KD embryos reached the blastocyst stage, similar to control embryos: 65.7% for control and 70.0% for MERVL-KD ($P=0.712$, chi-square test) with no evidence of developmental delay and morphological defects (Figure 3c, d), we concluded that retroviral proteins encoded by MERVL are dispensable for preimplantation development. These findings together show that genotoxicity from siRNA induction does not affect our main conclusion.

*Additionally, for studies that have been done with knockout mice in which transcriptional regulators of ZGA have been perturbed (such as *Dppa2/4*), were levels of MERVL assessed?*

We thank the reviewer for the comment. It has been reported that single depletion of either transcriptional regulators that associated with ZGA, such as *Dux*, *Dppa2/4* or *Obox4*, exhibited only minor or no changes in MERVL expression level. We now add this notion to the text (Page 18, Line 634~) with appropriate citations.

Minor concerns:

1. *Could the authors discuss more why MERVL begins to be transcribed at the zygote stage, but doesn't appear in the cytoplasm until the mid 2-cell stage (Fig. 1d)? Is there some sort of delay in RNA export from the nucleus at that developmental time?*

We agree that it is interesting to elucidate how MERVL transcript is retained in the nucleus of embryos at zygotes and early 2-cell stages. At this time, we speculate that MERVL RNA may interact with other RNA binding protein partners to reside in the nucleus. For example, there is evidence that transcript from LINE1 acts as a chromatin-associated RNA that binds Nucleolin, Kap1 and chromatin in regulating gene expression for the self-renewal of ES cells (Percharde, M. et al. 2018). Clarification of the potential mechanism for nuclear localization of MERVL RNA warrants our future investigation. We now added this notion to the text (Page 18, Line 641~) with appropriate citations.

2. *There seems to heterogeneity in MERVL levels (as determined by smFISH) between nuclei in the 4-cell embryo (Fig. 1d). Is this a general trend across all images examined, and if so, what might the consequences of such heterogeneity be?*

We thank the reviewer for the comment. According to the reviewer's suggestion, we focused on substantial variance in a transcriptional level of MERVL in each blastomere from 4-cell stage embryos. In ~50% of 4-cell stage embryos we examined (n = 5/9), there was at least one blastomere with high levels of MERVL transcript signal, suggesting that this cellular heterogeneity in MERVL expression contributes to functional consequences for full-term development of embryos. We now added this notion to the text (Page 4, Line 139~) with appropriate citations.

3. The key for Fig. 2d lists both 8-64 cell embryos as well as morula stage embryos. The morula stage begins at 16 cells, and thus these two classifications overlap with each other. Please update the legend (as well as the analyses) to follow this nomenclature.

Thank you. We have corrected them.

4. A major conclusion of this work is that the requirement for MERVL is due to either cis-acting properties, or the act of transcription of the transposable element. To make this conclusion, Sakashita et al. rely on comparisons between ASO-mediated knockdown, miRNA-mediated knockdown, and ASO-mediated knockdown with rescue of MERVL in trans. To strengthen these conclusions, it would be nice if the authors showed, for all treatments, both the MERVL RNA levels and localization (using smFISH) in addition to the MERVL Gag protein levels at the early and late 2-cell stage. This would allow readers to understand the extent to which cytoplasmic MERVL RNA levels are altered in each scenario.

We appreciate this important point and have provided data for smFISH against MERVL RNA at early and late 2-cell stages in each experimental condition (New Figure 2b and New Figure 3b, f, j).

5. For all figures showing genome browser tracks (Fig. 4i, Fig. 5d) for specific genes, some sort of scale bar or axis should be used to specify the relative heights of each of the tracks.

We thank the reviewer for the comment. We have added labels on the y-axis for the normalized read count, more clearly to representative track views (in Figure 4i and 5d, New Figure 6f, New Supplementary Figure S6f and S7c).

Reviewer #3:

The manuscript provides interesting and important new insights into the roles of endogenous retroviruses during early development. Endogenous retroviruses (ERVs) are prevalent ancient transposable elements in the mammalian genomes. Much of the importance of ERVs to cellular functions has been attributed to their Long Terminal Repeats (LTRs) encoding transcription factor binding sites, and acting as promoters and enhancers of cellular genes (e.g. Macfarlan et al., Nature 2012). This manuscript provides evidence that the RNA species produced at MERVL ERVs have critical regulatory roles in early murine embryos, shedding new light into how retrotransposons may affect critical cellular functions.

The data in the manuscript appear of high quality and appropriate for supporting the authors conclusions. I only have one major and a few minor suggestions for improvements.

We thank the reviewer for acknowledging the important contribution of our work to the field of transposon and early embryo development research. This is very important, we agree.

Major comment:

The main conclusions of the manuscript rely on ASO-mediated knockdown of MERVL RNA. The authors performed substantial bioinformatic analyses and control experiments to validate the ASOs they used. However, ASO-mediated knockdown is generally prone to artefacts since the pathways that lead to RNA degradation are incompletely understood. Since the key insights of the manuscript hinge on specificity of the ASO-knockdown, further controls or an alternative approach is highly recommended. For example, the authors could design and test various other sets of ASOs that target non-overlapping regions and test the existing and new ASOs against a larger set of control ASO including more scrambled controls. Also, the CasRX system (Konnerman et al., Cell 2018) could be used as an ASO-independent knockdown approach that works in cis to validate the ASO-results. shRNAs are also known to be processed and mediate knockdown in the nucleus to some degree.

We thank the reviewer for his/her careful consideration of our manuscript and for constructive criticisms and suggestions. Because we only relied on ASO-mediated KD to show biological significance of MERVL transcript during transition phase from totipotency to pluripotency in our original version of manuscript, the reviewer's invaluable comment led us to newly perform (1) microinjection of SOs which are complementary to the individual MERVL-targeting ASO, as a control for ASO-mediated KD, (2) CasRx-mediated KD and (3) CRISPR-mediated repression of MERVL, as the alternative approaches to prove the robustness of embryonic phenotype upon MERVL-KD.

In this revised manuscript, we perform experiments to exclude the possibility that the phenotypes of ASO-mediated MERVL-KD embryos arise from an artificial effect by ASO injection. As mentioned above, we have performed microinjection of SOs that were complementary to respective ASOs against MERVL as a control and shown no discernible defects in SO-injected embryos at the expression level of MERVL transcript and developmental competence (New Supplementary Figure S2), indicating that the phenotype of ASO-mediated MERVL-KD embryos is unlikely to arise from artifacts such as DNA replication stress and innate immune response.

Next, we tested KD of MERVL using the CasRx system with three-independent mature gRNAs harboring same targeting sequences as respective ASOs against MERVL. As a result, we

confirmed that the developmental rate to the blastocyst stage was significantly reduced upon CasRx-mediated MERVL-KD (New Supplementary Figure S3), although cytoplasmic and nuclear MERVL transcripts had not been fully knocked down by the CasRx-mediated system, unlike after induction of ASOs.

Finally, we also have performed CRISPRi-mediated repression of MERVL with dCas9-KRAB-MeCP2 and multiple-targeting gRNA constructs and revealed that MERVLi embryos partially mimicked ASO-mediated KD of MERVL with regard to developmental defects and retaining 2C transcriptome, although in which expression of MERVL was not fully repressed, probably due to inefficient expression of dCas9-KRAB-MeCP2 at zygotes-to-early 2-cell stages in our system (New Figure 3 and New Supplementary Figure S6). Taken together, although ASO-mediated KD is generally prone to artifacts and the pathways that lead to RNA degradation are incompletely understood, our findings based on the use of the above-mentioned strategies thoroughly corroborates that observed phenotypes in MERVL-KD embryos using ASOs are robust and not relevant to processing artifacts by ASO injection.

We have added these new data to the text (Page 5, Line 174~; Page 6, Line 196~; Page 8, Line 281~; Page 11, Line 391~; Page 17, Line 588~ and 596~) and figures (New Figure 3 and New Supplementary Figure S2, S3 and S6) with appropriate citations.

Minor comments:

1. The authors call MERVL RNA species (and RNA FISH assay) “nascent”, but then explain that MERVL RNAs are intronless. Since the term ‘nascent’ typically refers to unprocessed, intron-containing RNAs, clarification of the terminology is recommended.

We thank the reviewer for pointing this out. We have corrected this term to “MERVL transcript”.

2. Some of the transcriptomic changes highlighted by the authors seem minor. To help readers gauge extent, please add y-axes with units to the displayed Seq data (e.g. Fig 4i, Fig 5d, Supplementary Fig 5c).

We thank the reviewer for the comment. We have added labels on the y-axis for the normalized read count, more clearly to representative track views (in Figure 4i and 5d, New Figure 6f, New Supplementary Figure S6f and S7c).

Decision Letter, first revision:

11th Nov 2022

Dear Haru,

Thank you for submitting your revised manuscript entitled "Transcription of Murine Endogenous Retrovirus MERVL Is Required for Progression of Development in Early Preimplantation Embryos" (NG-A59732R). It has now been seen by reviewers #1 and #3 and their comments are below. Unfortunately, despite our multiple chase emails, reviewer #2 has not submitted a timely report. We have now decided to proceed with a decision.

The reviewers find that the paper has improved in revision, and therefore we'll be happy in principle to publish it in Nature Genetics, pending minor revisions to satisfy the referees' final requests and to comply with our editorial and formatting guidelines.

Thank you again for your interest in Nature Genetics. Please do not hesitate to contact me if you have any questions.

Congratulations!

Sincerely,

Tiago

Tiago Faial, PhD
Chief Editor
Nature Genetics
<https://orcid.org/0000-0003-0864-1200>

Reviewer #1 (Remarks to the Author):

I stand by my original review that this is incredibly important work for the field and commend the authors for the amount of new data and analyses included in this impressive revision.

The authors have addressed the main two criticisms that I raised in the initial submission.

1. Regarding the specificity of the ASO technique, they have included an additional control of the SO oligo which is a more convincing negative control to compare their MERVL ASO to. They have included an additional set of experiments with Crispr-I technology to repress transcription of the MERVL

elements (Fig 3i, j,kl). The Crispr-I experiments also show a similar trend (although less dramatic difference compared to the control) of the ASO experiments presented in 2d.

2. The other major criticism (a plausible explanation for how the molecular mechanism of MERVL ASO inhibits development) is less convincingly addressed. However, they provide several lines of evidence: p53 activation, p53-target gene perdurance, transcriptional profiling showing elevated 2-Cell transcripts outside of the normal ZGA window, global chromatin differences between control and MERVL ASO embryos, etc. And importantly, in the revised manuscript they find (Fig 6a) that unannotated transcripts near MERVL (uTAR) are significantly downregulated in MERVL ASO treated embryos. In these situations where a cumulative dysregulation of the transcriptome/epigenome results in decreased developmental potential it can be difficult to explain except for the "death by a thousand cuts" model. I would have loved a more tidy model, but the importance of the phenomenology and findings presented here warrant publication.

Minor criticisms: sufficiently addressed.

Additional comments on the revision: it appears that most of the figure legends do not include a reference the sample size. Typically, it is Nature Genetics format to state the n= as number of embryos, or number of independent experiments performed. Additionally, for critical developmental analyses like Fig 2d, Fig 3d, 3h, 3i it is unclear if this is only meant to be qualitative data or any statistical comparisons are drawn. No statistical test is referenced in 2d, 3d, 3h, 3i and the sample size (n= number of embryos) or the number of times these experiments were executed are not included in the figure legends.

Reviewer #3 (Remarks to the Author):

The authors have addressed my comments appropriately. The additional data using sense oligos (SO), CRISPRi and CasRX to knock down MERVL provide important support for the main insights of the study.

I have a few minor comments on the newly added data and text.

1. Regarding the CasRX experiments, could the authors clarify in the text what exactly in the data is interpreted in the following statement: "although the developmental competence of embryos reaching blastocyst stage was significantly reduced CasRx-mediated MERVL-KD" (line 207)? The CasRX embryos have similar stage profiles as the controls, and it is unclear what numbers are interpreted in the above statement.

2. I encourage the authors to add the number of embryos analyzed in Figure 3, not just the percentages. Mentioning in the text how many embryos were analyzed in how many injection experiments would also help readers better appreciate these data.

3. Please check and adjust the wording in the sentence "loss of MERVL transcript led to retain an accessible chromatin state at, and aberrant expression of, a subset of 2-cell specific genes" (line 33). (e.g. "retention" instead of "retain").

4. The term “highly confident” MERVL... appears a few times in the text. I assume the authors mean “high confidence MERVL” copies.

5. A recent paper in Nature Genetics reported that MERVK and IAP RNA can act in cis in early mouse embryos (Asimi et al, 2022). The authors are encouraged to mention that cis effects of ERV RNAs could thus be a ubiquitous feature of ERVs, not restricted to MERVL.

Author Rebuttal, first revision:

Point-by-Point RESPONSE TO REVIEWERS

Reviewer #1 (Remarks to the Author):

I stand by my original review that this is incredibly important work for the field and commend the authors for the amount of new data and analyses included in this impressive revision.

The authors have addressed the main two criticisms that I raised in the initial submission.

1. Regarding the specificity of the ASO technique, they have included an additional control of the SO oligo which is a more convincing negative control to compare their MERVL ASO to. They have included an additional set of experiments with Crispr-I technology to repress transcription of the MERVL elements (Fig 3i, j,kl). The Crispr-I experiments also show a similar trend (although less dramatic difference compared to the control) of the ASO experiments presented in 2d.

2. The other major criticism (a plausible explanation for how the molecular mechanism of MERVL ASO inhibits development) is less convincingly addressed. However, they provide several lines of evidence: p53 activation, p53-target gene perdurance, transcriptional profiling showing elevated 2-Cell transcripts outside of the normal ZGA window, global chromatin differences between control and MERVL ASO embryos, etc. And importantly, in the revised manuscript they find (Fig 6a) that unannotated transcripts near MERVL (uTAR) are significantly downregulated in MERVL ASO treated embryos. In these situations where a cumulative dysregulation of the transcriptome/epigenome results in decreased developmental potential it can be difficult to explain except for the "death by a thousand cuts" model. I would have loved a more tidy model, but the importance of the phenomenology and findings presented here warrant publication.

We would like to thank the reviewer for acknowledging the important contribution of our work to the field of transposon and early embryo development research.

Minor criticisms: sufficiently addressed.

Additional comments on the revision: it appears that most of the figure legends do not include a reference the sample size. Typically, it is Nature Genetics format to state the n= as number of embryos,

or number of independent experiments performed. Additionally, for critical developmental analyses like Fig 2d, Fig 3d, 3h, 3i it is unclear if this is only meant to be qualitative data or any statistical comparisons are drawn. No statistical test is referenced in 2d, 3d, 3h, 3i and the sample size (n= number of embryos) or the number of times these experiments were executed are not included in the figure legends.

Thank you for pointing out these preparation problems. Accordingly, we have edited and corrected figure legends.

Reviewer #3 (Remarks to the Author):

The authors have addressed my comments appropriately. The additional data using sense oligos (SO), CRISPRi and CasRX to knock down MERVL provide important support for the main insights of the study.

We would like to thank the reviewer for acknowledging the important contribution of our work to the field of transposon and early embryo development research.

I have a few minor comments on the newly added data and text.

1. Regarding the CasRX experiments, could the authors clarify in the text what exactly in the data is interpreted in the following statement: “although the developmental competence of embryos reaching blastocyst stage was significantly reduced CasRx-mediated MERVL-KD” (line 207)? The CasRX embryos have similar stage profiles as the controls, and it is unclear what numbers are interpreted in the above statement.

We thank the reviewer for his/her careful consideration of our manuscript. We have indicated the numbers of embryos we examined and statistical difference in developmental rate between control and CasRx-mediated KD embryos, in Figure legend of Extended Data Figure 3. Due to the limitation of words numbers for publication, we’ve edited above sentence, appropriately.

2. I encourage the authors to add the number of embryos analyzed in Figure 3, not just the percentages. Mentioning in the text how many embryos were analyzed in how many injection experiments would also help readers better appreciate these data.

We thank the reviewer for his/her careful consideration of our manuscript. We have indicated the numbers of embryos we examined.

3. Please check and adjust the wording in the sentence “loss of MERVL transcript led to retain an accessible chromatin state at, and aberrant expression of, a subset of 2-cell specific genes” (line 33). (e.g.

“retention” instead of “retain”).

Thank you. We have checked the wording and accordingly edited the sentence.

4. The term “highly confident” MERVL... appears a few times in the text. I assume the authors mean “high confidence MERVL” copies.

Thank you. We have corrected accordingly.

5. A recent paper in Nature Genetics reported that MERVK and IAP RNA can act in cis in early mouse embryos (Asimi et al, 2022). The authors are encouraged to mention that cis effects of ERV RNAs could thus be a ubiquitous feature of ERVs, not restricted to MERVL.

Thank you. We have cited the paper and added the following sentence to “Discussion” section; “Likewise, there has been reported that the transcribed RNA from other type of ERVs (i.e, IAPs and MMERVK10C) can act in cis in ESC (ref : Asimi V. et al. Nat Genet. 2022).”

Final Decision Letter:

26th Jan 2023

Dear Haru,

I am delighted to say that your manuscript "Transcription of MERVL Retrotransposons Is Required for Preimplantation Embryo Development" has been accepted for publication in an upcoming issue of Nature Genetics.

Your paper will be published online after we receive your corrections and will appear in print in the

next available issue. You can find out your date of online publication by contacting the Nature Press Office (press@nature.com) after sending your e-proof corrections. Now is the time to inform your Public Relations or Press Office about your paper, as they might be interested in promoting its publication. This will allow them time to prepare an accurate and satisfactory press release. Include your manuscript tracking number (NG-A59732R1) and the name of the journal, which they will need when they contact our Press Office.

Please note that *Nature Genetics* is a Transformative Journal (TJ). Authors may publish their research with us through the traditional subscription access route or make their paper immediately open access through payment of an article-processing charge (APC). Authors will not be required to make a final decision about access to their article until it has been accepted. [Find out more about Transformative Journals](https://www.springernature.com/gp/open-research/transformative-journals)

Authors may need to take specific actions to achieve [compliance with funder and institutional open access mandates](https://www.springernature.com/gp/open-research/funding/policy-compliance-faqs). If your research is supported by a funder that requires immediate open access (e.g. according to [Plan S principles](https://www.springernature.com/gp/open-research/plan-s-compliance)) then you should select the gold OA route, and we will direct you to the compliant route where possible. For authors selecting the subscription publication route, the journal's standard licensing terms will need to be accepted, including [self-archiving-and-license-to-publish](https://www.nature.com/nature-portfolio/editorial-policies/self-archiving-and-license-to-publish). Those licensing terms will supersede any other terms that the author or any third party may assert apply to any version of the manuscript.

Please note that Nature Portfolio offers an immediate open access option only for papers that were first submitted after 1 January, 2021.

To assist our authors in disseminating their research to the broader community, our SharedIt initiative provides you with a unique shareable link that will allow anyone (with or without a subscription) to

read the published article. Recipients of the link with a subscription will also be able to download and print the PDF.

An online order form for reprints of your paper is available at https://www.nature.com/reprints/author-reprints.html. Please let your coauthors and your institutions' public affairs office know that they are also welcome to order reprints by this method.

Sincerely,

Tiago

Tiago Faial, PhD
Chief Editor
Nature Genetics
<https://orcid.org/0000-0003-0864-1200>